# Learning to stand with sensorimotor delays generalizes across directions and from hand to leg effectors
Brandon G. Rasman [1,2,3], Jean-Sébastien Blouin[4,5,6], Amin M. Nasrabadi [4], Remco van Woerkom[1], Maarten A. Frens[1] & Patrick A. Forbes [1]✉

Humans receive sensory information from the past, requiring the brain to overcome delays to perform daily motor skills such as standing upright. Because delays vary throughout the body and change over a lifetime, it would be advantageous to generalize learned control policies of balancing with delays across contexts. However, not all forms of learning generalize. Here, we use a robotic simulator to impose delays into human balance. When delays are imposed in one direction of standing, participants are initially unstable but relearn to balance by reducing the variability of their motor actions and transfer balance improvements to untrained directions. Upon returning to normal standing, aftereffects from learning are observed as small oscillations in control, yet they do not destabilize balance. Remarkably, when participants train to balance with delays using their hand, learning transfers to standing with the legs. Our findings establish that humans use experience to broadly update their neural control to balance with delays.

Delays are a ubiquitous feature of movement control. Consequently, the nervous system of all animals must learn to control movement based on outdated sensory information and compensate for the influence delays have on self-motion. The delays accompanying the control of different motor effectors, however, are not constant; instead, they vary due to the length and velocity of neural transmission[1,2] and the complexity of the neural networks involved[1–6]. Furthermore, delays can lengthen throughout the lifespan due to alterations in nerve conduction, muscle force generation, and neural processing associated with growth[3], aging[7], and disease[8,9]. Given the likelihood that sensorimotor delays can change[3,7,10–12], it would be advantageous if the brain could generalize learned policies for controlling self-motion with delays and transfer them to different contexts. Standing balance represents a motor behavior that would largely benefit from generalizing learned control with delays, given that the human bipedal posture is mechanically unstable[13–16] and failing to accommodate for different sensorimotor delays in balance control increases the risk of falling[17–20]. However, the generation of multidirectional balance-correcting responses relies on a diversity of sensory signals (i.e., visual, vestibular, somatosensory, auditory) and anatomically distinct muscles (i.e., ankle and hip), which may challenge and limit generalization across directions and muscle effectors.

Imposing long delays into anteroposterior control of human standing destabilizes balance, but through training, participants can learn to regain their upright balance control and retain this ability after three months[21]. This learning is accompanied by modulations in the vestibular control of balance and perception of self-motion, supporting the view that adaptations to ongoing balance control are governed through sensorimotor processes that can change our perceptual awareness of ongoing balance[22–26]. However, whether the nervous system can generalize the learning to different task contexts remains unknown. Typically, the more contextual factors that overlap between tasks (i.e., goal, movement mechanics, sensory cues, and motor effectors), the more likely learned control policies will generalize[27–30]. Standing balance involves controlling whole-body motion in both anteroposterior and mediolateral space, with each direction of balance possessing distinct biomechanics[16,17,31,32], muscle effectors, activation patterns[33–35], and sensorimotor delays[5,17,20,36,37]. Therefore, these differing sensorimotor factors may limit the ability to generalize learning across orthogonal space ($H_0$, see Fig. 1), as observed when standing participants adapt their balance to externally imposed perturbations in different directions[38]. On the other hand, the common task goal of balancing the upright body against gravity may help facilitate transfer across the different directions of balance ($H_1$).

[1]Department of Neuroscience, Erasmus MC, University Medical Center Rotterdam, Rotterdam, The Netherlands. [2]School of Physical Education, Sport and Exercise Sciences, University of Otago, Dunedin, New Zealand. [3]Donders Institute for Brain, Cognition and Behaviour, Radboud University, Nijmegen, The Netherlands. [4]School of Kinesiology, University of British Columbia, Vancouver, BC, Canada. [5]Djavad Mowafaghian Centre for Brain Health, University of British Columbia, Vancouver, BC, Canada. [6]Institute for Computing, Information and Cognitive Systems, University of British Columbia, Vancouver, BC, Canada. ✉e-mail: p.forbes@erasmusmc.nl

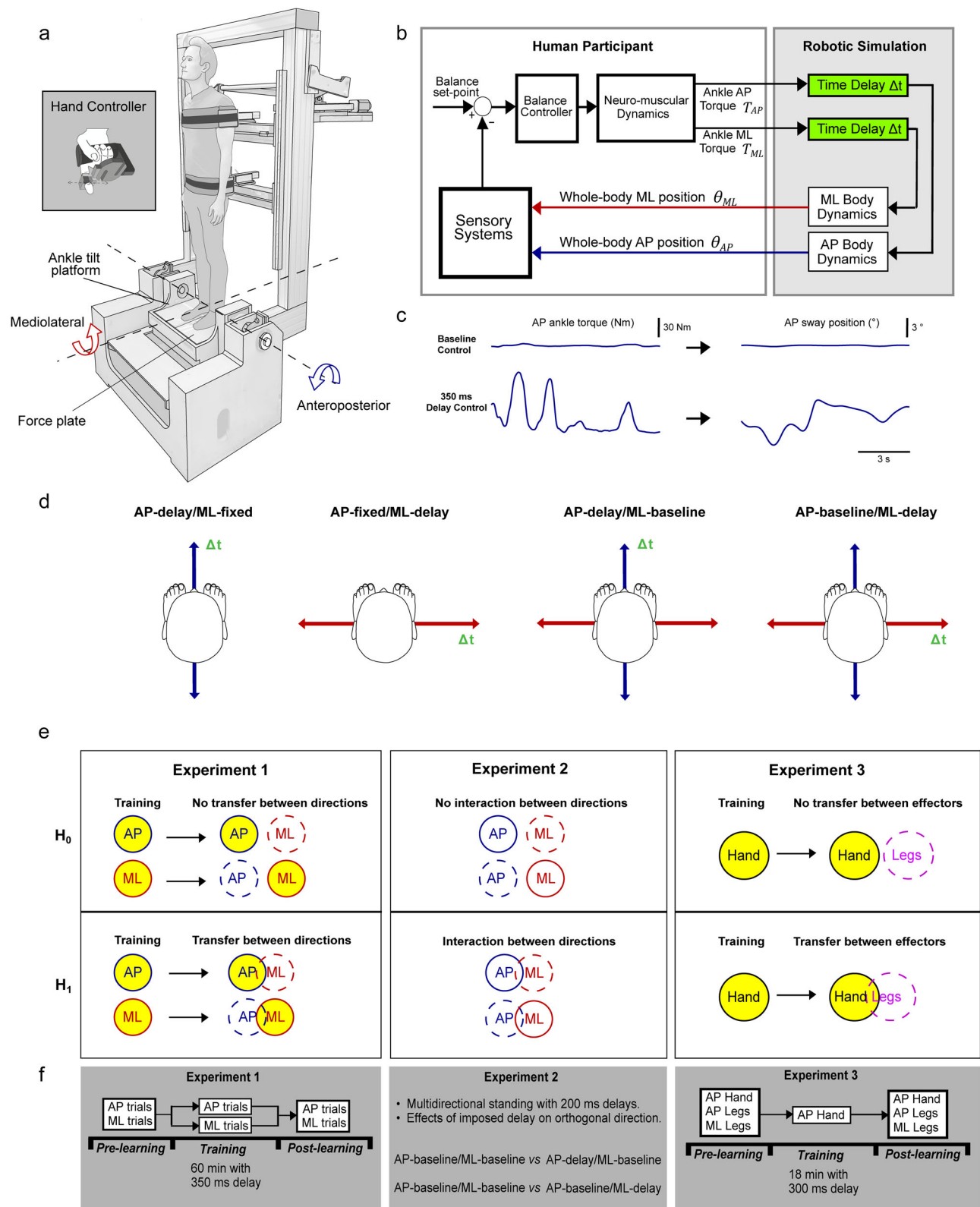

Here, we explored these two hypotheses to determine whether the learned control policies of balancing with sensorimotor delays generalize across different contexts.

An important consideration for any potential generalization is that lower limb muscles generate joint forces and torques contributing to both balance directions[39–42]. As a result, the question arises of whether any transfer of balance improvements is due to a neural generalization of learning or a byproduct of biomechanical interactions (i.e., adapted motor commands of muscles contributing to multidirectional balance control). Importantly, the possibility of neural generalization and biomechanical interactions are not mutually exclusive – i.e., both could contribute to the transfer of training benefits. Therefore, to determine whether biomechanical

**Fig. 1 | Experimental set-up and hypothesized outcomes. a** Participants stood on a force plate mounted to an ankle-tilt platform and were securely strapped to the robot frame with torso and pelvis harnesses. In conditions where whole-body motion was controlled by forces and torques produced at the feet, the support surface was held horizontally (earth-fixed reference) while the robot moved the participant's whole body in the anteroposterior and/or mediolateral directions. In conditions where participants controlled anteroposterior whole-body motion by modulating force at their right index finger (see inset: hand controller), the support surface co-rotated with the backboard (i.e., ankle sway referencing) to remove the sensory conflict that arises from ankle somatosensory feedback not being coupled to ankle-generated torques to control balance (see "Methods"). **b** Participants balanced the robotic simulator as it operated with different delays (4–350 ms). Delays were imposed in either a single direction or both directions of balance motion. Control signals from the legs (ankle-produced torque) or hand (torque generated at the finger) were buffered in the robotic simulation computer model such that angular rotation of the whole body could be delayed. **c** Sample raw data of ankle torque and center of mass (i.e., whole-body) angular position in baseline and 350 ms delay AP standing balance trials. **d** Across experiments, we presented conditions that restrained balance motion to a single direction while others necessitated balancing in both directions. Imposed

delays were added either to one or both directions of motion. **e** Hypothesized outcomes of the separate experiments. Experiment 1: training to balance with delays in a single direction will result in balance improvements in only the trained direction ($H_0$) or in both directions ($H_1$). Experiment 2: imposed delays targeting a single direction of motion will destabilize standing only in that direction ($H_0$) or in both directions ($H_1$). Experiment 3: training to balance with delays during the hand-control condition will result in balance improvements only for balance control with the hand ($H_0$) or will transfer to balance control with the legs ($H_1$). Yellow circles indicate conditions where delayed balance was learned (i.e., in specific directions or with specific motor effectors). **f** Experimental design and timelines. In Experiment 1, one group trained in the AP-delay/ML-fixed condition while another trained in the AP-fixed/ML-delay condition (see Methods). In Experiment 2, a new group of participants performed short (20-second) standing trials with different combinations of balance direction and delays. In Experiment 3, a new group of participants trained to balance with a 300 ms imposed delay while controlling whole-body motion in the AP direction using a hand controller. Elements in part (**a**) are adapted with permission from[79], Frontiers Media SA. Elements in part (**d**) are adapted with permission from[69], JNeurosci.

interactions are required for the transfer of training benefits, we assessed whether transfer of learning occurred across biomechanically independent muscles. If the brain can broadly update its control to accommodate for imposed delays, we hypothesized that even when participants trained to balance the whole body with their hand muscles, balance performance would improve when controlling the upright body with postural lower limb muscles.

We performed three experiments in which participants stood in a robotic balance simulator (Fig. 1) that recreates the physical sensations and neural signals for balancing upright, while delays were imposed into the anteroposterior and/or mediolateral directions of balance. In Experiment 1, participants practiced balancing with an imposed delay in one direction and were then asked to balance with delays in both the trained and untrained directions. We found that regardless of the direction trained (anteroposterior or mediolateral), participants improved their balance (reduced movement variability and increased balancing time) across orthogonal space. In Experiment 2, participants balanced while delays were imposed for brief periods (20 s) in one or both the anteroposterior and mediolateral directions. We found evidence of biomechanical interactions that may contribute to the transfer observed in Experiment 1 because imposing a delay in a single direction increased the variability of standing movement in both the delayed and orthogonal (non-delayed) directions. In our final experiment (Experiment 3), we tested whether the observed transfer of balance learning (i.e., Experiment 1) relied on the biomechanical interactions identified in Experiment 2. Here, we designed a condition that allowed whole-body motion to be controlled through hand-generated forces. A new group of participants were trained to balance with a delay in this hand control condition and were evaluated for any transfer of learning to balance with a delay using their leg muscles. After training, participants improved their balance in both the trained hand-controlled and untrained leg-controlled conditions. In all training experiments, debrief sessions indicated that participants deliberately learned to make calmer and smaller motor actions. Collectively, our findings demonstrate that humans can transfer the learned ability to balance upright with imposed delays across orthogonal space and biomechanically independent muscle effectors engaged in balance control. These results reveal that the human brain can leverage prior experience of balancing with sensorimotor delays to control upright posture in distinct contexts.

## Results

### Experiment 1: Learning to balance with imposed sensorimotor delays transfers across directions

Participants were instructed to stand quietly on a robotic balance simulator while a 350 ms delay was imposed between their ankle torques and consequent whole-body position (see "Methods"). Whole-body (i.e., the center

of mass) angular position, angular velocity, and ankle-generated torques were recorded to quantify standing behavior when balancing with or without the imposed delay. The robot was programmed to simulate whole-body balance in either (or both) the anteroposterior (AP) direction and/or the mediolateral (ML) direction. We programmed angular position limits of 6° anterior and 3° posterior for AP and 3° left/right for ML balance to represent the limits of whole-body position during standing (see "Methods").

Here, two different groups of participants trained to balance with imposed delays of 350 ms in either the AP ($n = 12$) or ML ($n = 12$) direction while the orthogonal direction was fixed. When imposing delays prior to learning, whole-body oscillations in both groups were highly variable and participants had difficulty remaining upright within the simulated limits (see representative data in Fig. 2a, top row). As a combined group ($n = 24$), participants oscillated with large angular velocity variance ($17.4 \pm 1.7$ (°/s)$^2$ in AP-delay trials; $12.9 \pm 1.2$ (°/s)$^2$ in ML-delay trials) and spent only ~63–64% of the time within the simulated balance limits (63% ± 1% of the time in AP-delay trials; 64% ± 1% in ML-delay trials).

**Training improves balancing with delays.** During training, participants in both groups progressively reduced the variability of their angular velocity and increased the percentage of time they maintained balance within the virtual limits (Fig. 2c). By the end of training, 10 out of 12 participants in the AP group and 9 out of 12 participants in the ML group could balance for at least 60 s without reaching the simulated limits. First-order-exponential fits to the angular velocity variance and percent time within the limits for both the AP and ML groups indicated that there was no difference in time constants between the two training groups for either angular velocity variance (AP training: $14.7 \pm 2.6$ min vs ML training: $13.7$ min ± $3.6$ min, $t_{(22)} = 0.22$, $p = 0.83$) or percent time within the balance limits (AP training: $18.0 \pm 2.7$ min vs ML training: $16.8$ min ± $2.2$ min, $t_{(22)} = 0.33$, $p = 0.74$). Similar outcomes were also observed in ankle-generated torque, which progressively decreased over the course of training (Fig. 3c). First-order-exponential fits to ankle torque SD for both the AP and ML groups indicated that there was no difference in time constants between the two training groups (AP training: $26.0 \pm 4.9$ min vs ML training: $19.6$ min ± $4.5$ min, $t_{(22)} = 0.97$, $p = 0.34$). Collectively, the similar time constants of all metrics between groups suggest that balance learning with the delay in AP and ML directions occurred at similar rates.

To assess the control underlying the learning process, we further examined the frequency content of angular velocity and ankle torque signals at the start and end of training. The auto spectra of both signals showed a general decrease in power across frequencies (Figs. 2d, 3d), aligning with the general reductions in variability described above. However, when we normalized the auto spectra to the sum of the power from 0.2–5 Hz, we

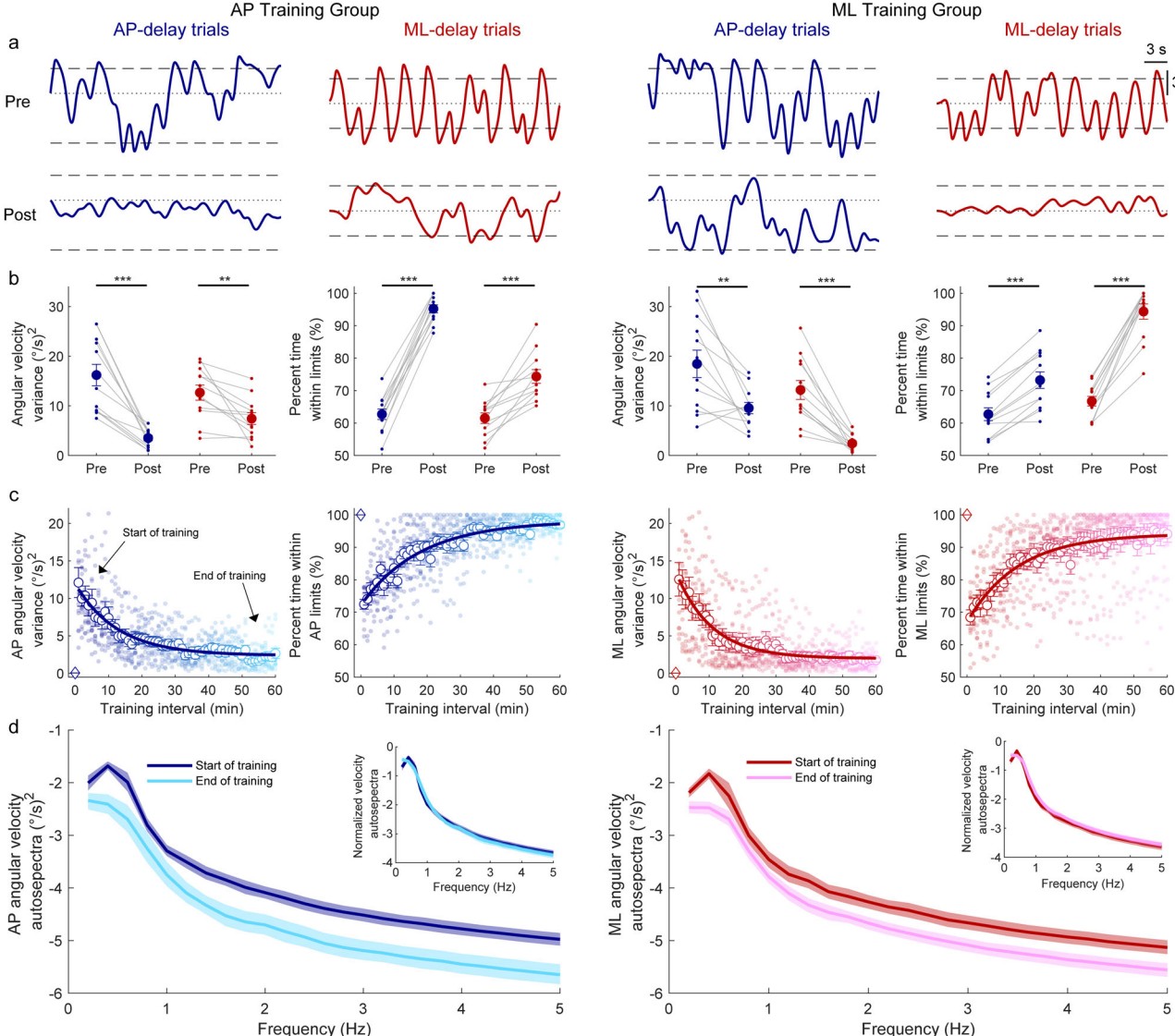

**Fig. 2 | Whole-body movement behavior with imposed delay from Experiment 1.**
**a** Whole-body (i.e., center of mass) angular position (°) from representative participants in the AP training group and ML training group. The top and bottom rows illustrate pre- and post-learning trials, respectively. During delayed standing trials, the robotic simulator operated with a 350 ms delay and whole-body motion was restricted such that motion could only occur in the AP or ML direction. Dashed lines represent the virtual position limits for AP (6° anterior, 3° posterior) and ML (3° left, 3° right) directions. Dotted lines represent the 0° position for all conditions.
**b** Angular velocity variance and percent time within the limits in the pre- and post-learning trials from both groups. Small circles connected with thin lines are individual participants and larger filled circles are group averages ($n = 12$ for each group) with SEM error bars. Regardless of which direction was trained (AP-delay or ML-delay), balance improvements were observed in both angular velocity variance and percent time within the limits for the AP and ML conditions. ** indicates $p < 0.01$ and *** indicates $p < 0.001$. **c** Single participant and average angular velocity

variance and percent time within the limits estimated over 1-minute intervals during the delay training trials from all participants who completed the training protocol (AP training group: $n = 12$; ML training group: $n = 12$). The color gradient from dark to light indicates the progression of training. The solid lines show fitting of average angular velocity variance and average percent time within the balance limits to a first-order exponential function using a least-square method:
$f(x) = a * \exp\left(\frac{-x}{b}\right) + c$. Data for one minute standing at baseline (system delay = 4 ms) are represented by open diamonds. Error bars are SEMs. **d** Group mean $\log_{10}$ transformed auto spectra of angular velocity signals extracted from the start and end of learning. Insets present auto spectra normalized to the sum of the power from 0.2–5 Hz, demonstrating a similar distribution of frequencies from the start and end of training. Shaded regions around the means represent the bootstrapped 95% confidence interval. For all panels, data in blue represent AP-delay trials whereas data in red represent ML-delay trials.

observed overlapping power spectra at the start and end of training (see insets Figs. 2d, 3d). This suggests that while the power of the controlling torque produced by participants decreased during learning, the distribution of frequencies remained consistent throughout.

**Transfer from training.** When participants balanced with the delay after the training, we saw improvements in balance in the trained direction. Planned comparisons (paired *t*-tests, Bonferroni corrected) of the pre- and post-learning trials (Figs. 2a, 3a for sample raw data, Figs. 2b, 3b for

group data) indicated that for all groups and in each direction of imposed delay, angular velocity variance and torque SD decreased, while percent time within the limits increased. Specifically, from pre- to post-learning trials, the twelve participants in the AP training group reduced AP angular velocity variance by ~78% (pre: 16.2 ± 2.2 (°/s)$^2$ vs post: 3.5 ± 0.5 (°/s)$^2$; $p < 0.001$), increased percent time within the AP limits (pre: 64% ± 2% vs. post: 95% ± 1%; $p < 0.001$), and reduced AP torque SD by ~55% (pre: 37.0 ± 4.0 Nm vs post: 16.6 ± 1.7 Nm; $p < 0.001$). Similarly, the twelve participants in the ML training group reduced ML angular

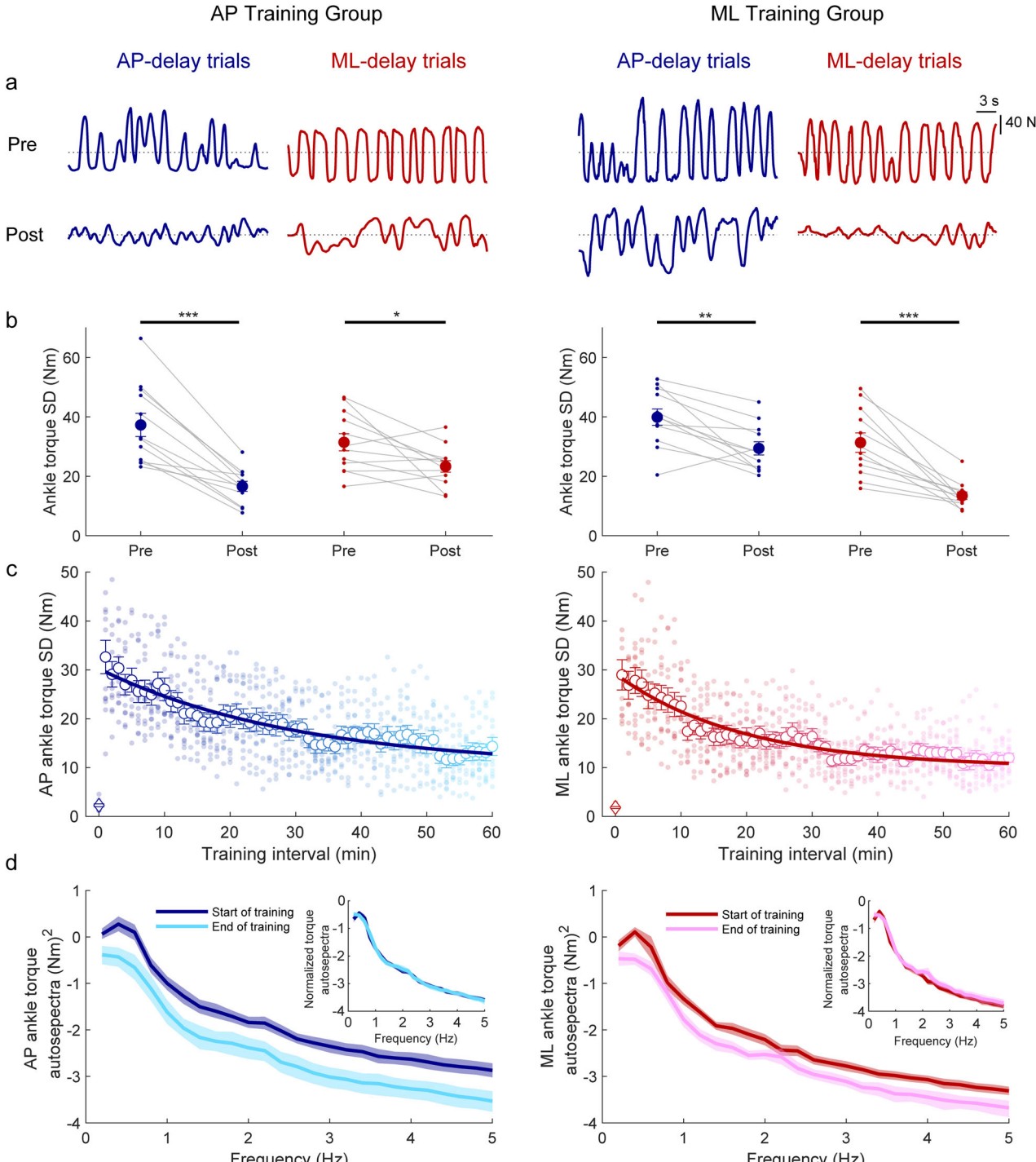

**Fig. 3 | Ankle-produced torque behavior with imposed delay from Experiment 1.**
**a** Ankle-produced torque (Nm) from representative participants in the AP training group and ML training group. The top and bottom rows illustrate pre- and post-learning trials, respectively. During delayed standing trials, the robotic simulator operated with a 350 ms delay, and whole-body motion was restricted to occur in only the AP or ML direction in response to AP or ML ankle torques, respectively. **b** Ankle torque standard deviation in the pre- and post-learning trials from both groups. Small circles connected with thin lines are individual participants and larger filled circles are group averages ($n = 12$ for each group) with SEM error bars. Regardless of the direction trained (AP-delay or ML-delay), ankle torque variability decreased for the AP and ML conditions. For all panels, data in blue represent AP-delay trials (and AP torque) whereas data in red represent ML-delay trials (and ML torque). * indicates $p < 0.05$, ** indicates $p < 0.01$ and *** indicates $p < 0.001$. **c** Single

participant and average ankle torque standard deviations estimated over 1-minute intervals during the delay training trials (AP training group: $n = 12$; ML training group: $n = 12$). The color gradient from dark to light indicates the progression of training. The solid lines show the fitting of average ankle torque standard deviation to a first-order exponential function using a least-square method:
$f(x) = a * \exp\left(\frac{-x}{b}\right) + c$. Data for one minute standing at baseline (system delay = 4 ms) are represented by open diamonds. Error bars are SEMs. **d** Group mean $\log_{10}$ transformed auto spectra of ankle torque signals extracted from the start and end of learning. Insets present auto spectra normalized to the sum of the power from 0.2–5 Hz, demonstrating the similar distribution of frequencies from the start and end of training. Shaded regions around the means represent the bootstrapped 95% confidence interval. For all panels, data in blue represent AP-delay trials whereas data in red represent ML-delay trials.

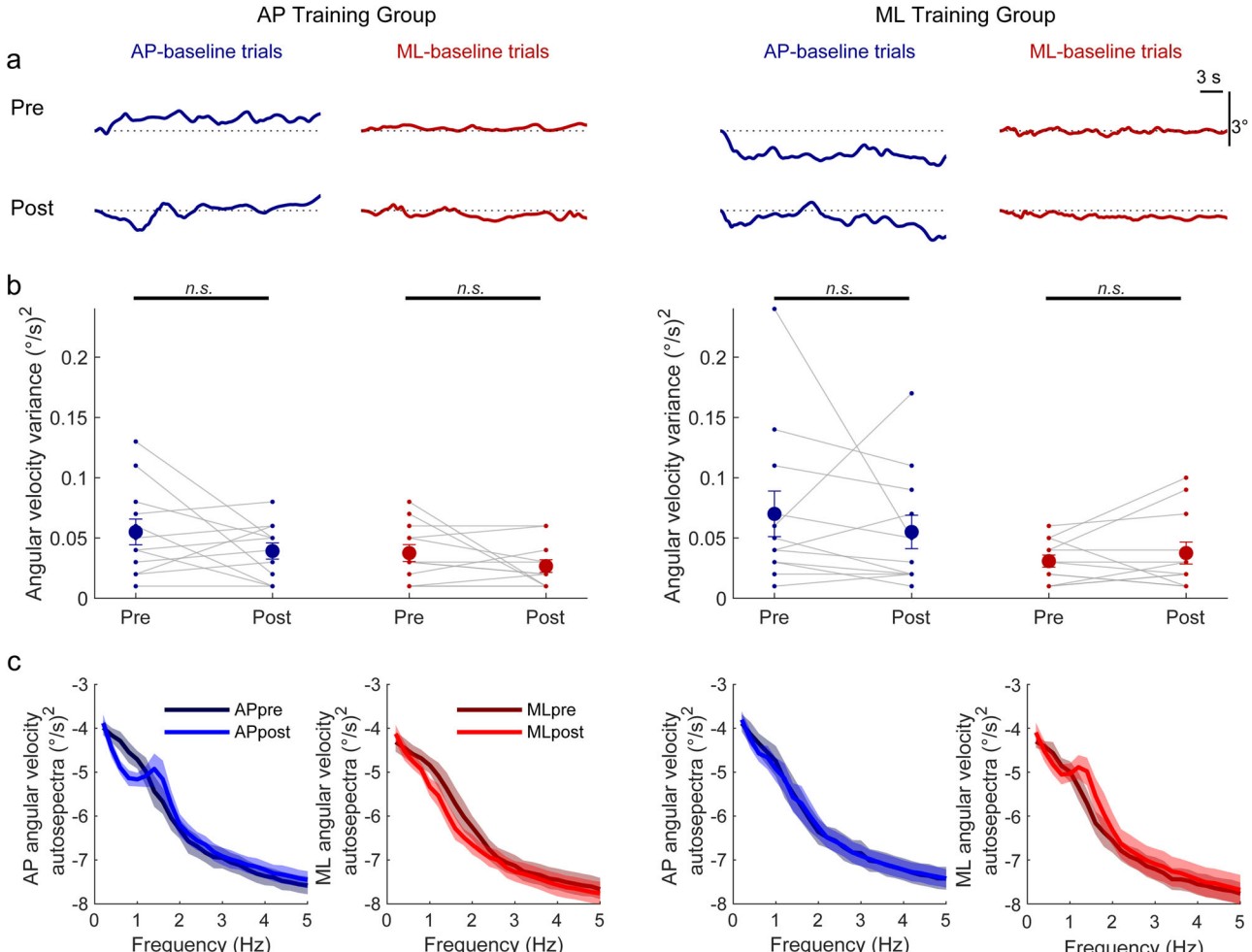

**Fig. 4 | Whole-body movement behavior in baseline (4 ms delay) trials from Experiment 1. a** Whole-body angular position (°) from representative participants in the AP training group and ML training group. The top and bottom rows illustrate pre- and post-learning baseline trials, respectively. Dotted lines represent the 0° position for all conditions. **b** Angular velocity variance in the pre- and post-learning AP-baseline and ML-baseline trials. Small circles connected with thin lines are individual participants and larger filled circles are group averages ($n = 12$ for each group) with accompanying SEM error bars. No changes were observed between pre- and post-learning baseline trials in angular velocity variance. Every participant balanced within the limits for 100% of the time for every baseline trial (not plotted). Not significant is indicated by n.s. **c** Group mean $\log_{10}$ transformed auto spectra of angular velocity signals extracted from pre- and post-baseline trials show changes in the distribution of power as a decrease at low frequencies (0.2–1.2 Hz) and a peak emerging at 1.4 Hz, primarily in the trained direction. Darker lines represent the mean of pre-learning trials, while lighter lines represent the mean of post-learning trials. Shaded regions around the means represent the bootstrapped 95% confidence interval. For all panels, data in blue represent AP-baseline trials whereas data in red represent ML-baseline trials.

velocity variance by ~82% (pre: $13.2 \pm 1.9$ $(°/s)^2$ vs post: $2.4 \pm 0.5$ $(°/s)^2$; $p < 0.001$), increased percent time within the ML limits (pre: $67\% \pm 1\%$ vs post: $94\% \pm 2\%$; $p < 0.001$) and reduced ML torque SD by ~57% (pre: $31.3 \pm 3.2$ Nm vs post: $13.5 \pm 1.3$ Nm; $p < 0.001$).

When participants were exposed to the delay in the untrained direction, while balance was locked in the trained direction, their balance performance also improved. Pairwise comparisons revealed that participants in the AP training group reduced ML angular velocity variance by ~42% (pre: $12.7 \pm 1.5$ $(°/s)^2$ vs post: $7.4 \pm 1.1$ $(°/s)^2$; $p = 0.004$), increased percent time within ML limits (pre: $61\% \pm 2\%$ vs. post: $74\% \pm 2\%$; $p < 0.001$) and reduced ML torque SD by ~26% (pre: $31.4 \pm 2.9$ Nm vs post: $23.2 \pm 1.9$; $p = 0.013$). Similarly, participants in ML training group reduced AP angular velocity variance by ~49% (pre: $18.5 \pm 2.8$ $(°/s)^2$ vs post: $9.5 \pm 1.1$ $(°/s)^2$; $p < 0.001$), increased their percent time within AP limits (pre: $63\% \pm 2\%$ vs post: $73\% \pm 3\%$; $p < 0.001$) and reduced AP torque SD by ~26% (pre: $39.9 \pm 2.8$ Nm vs post: $29.4 \pm 2.2$; $p = 0.001$). Further analysis demonstrated that while participants improved balance performance in both trained and untrained directions, relative improvements were greater in the trained direction (Supplementary Note 1, Supplementary Fig. 1). Importantly, a control experiment demonstrated that simply experiencing balance control

in the robot did not result in improvements between pre- and post-trials (see *Control experiment* below). Therefore, these results suggest that training to balance with a delay in one direction transfers to the orthogonal direction.

**Baseline standing before and after learning.** We further assessed whether learning to balance with the 350 ms delay resulted in any aftereffects in normal balance by comparing baseline trials (i.e., 4 ms delay) from pre- and post-learning sessions (Figs. 4, 5). In the post-learning baseline trials, angular velocity variability was similar to pre-learning trials and participants always remained within the virtual limits. In both AP and ML training groups, there was no difference in angular velocity variance (Fig. 4b) between the pre and post AP-baseline trials (AP training: $0.05 \pm 0.01$ vs $0.04 \pm 0.01$ $(°/s)^2$, $p = 0.19$; ML training: $0.07 \pm 0.02$ vs $0.06 \pm 0.01$ $(°/s)^2$, $p = 0.46$) and no difference in angular velocity variance between the pre and post ML-baseline trials (AP training: $0.04 \pm 0.01$ vs $0.03 \pm 0.01$ $(°/s)^2$, $p = 0.20$; ML training: $0.03 \pm 0.01$ vs $0.04 \pm 0.01$ $(°/s)^2$, $p = 0.38$). Analysis of ankle torque variability showed similar results (Fig. 5b). In both AP and ML training groups, there was no difference in AP torque SD between the pre and post AP-baseline trials (AP training: $2.29 \pm 0.29$ Nm vs $2.12 \pm 0.22$ Nm,

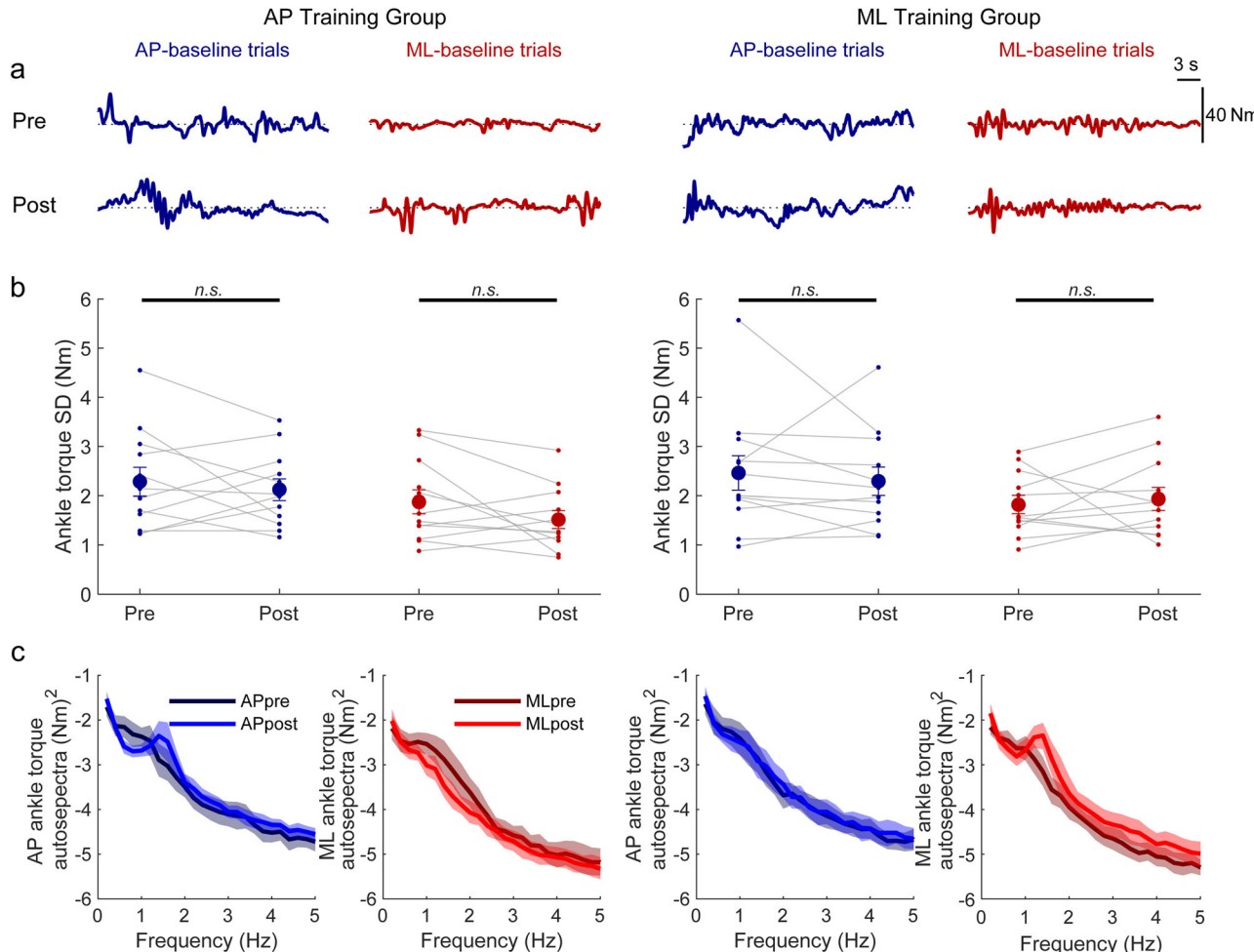

**Fig. 5 | Ankle-torque behavior in baseline (4 ms delay) trials from Experiment 1. a** Ankle torque (Nm) from representative participants in the AP training group and ML training group. The top and bottom rows illustrate pre- and post-learning baseline trials, respectively. Dotted lines represent 0 Nm for all conditions. **b** Ankle torque SD in the pre- and post-learning AP-baseline and ML-baseline trials. Small circles connected with thin lines are individual participants and larger filled circles are group averages ($n = 12$ for each group) with accompanying SEM error bars. No changes were observed between pre- and post-learning baseline trials in ankle torque standard deviation. Not significant is indicated by n.s. **c** Group mean $\log_{10}$ transformed auto spectra of ankle torque signals extracted from pre and post-baseline trials show changes in the distribution of power as a decrease at low frequencies (0.2–1.2 Hz) and a peak emerging at 1.4 Hz, primarily in the trained direction. Darker lines represent the mean of pre-learning trials, while lighter lines represent the mean of post-learning trials. Shaded regions around the means represent the bootstrapped 95% confidence interval. For all panels, data in blue represent AP-baseline trials whereas data in red represent ML-baseline trials.

$p = 0.52$; ML training: $2.46 \pm 0.35$ Nm vs $2.29 \pm 0.29$ Nm, $p = 0.57$) and no difference in ML torque SD between the pre and post ML-baseline trials (AP training: $1.88 \pm 0.24$ Nm vs $1.52 \pm 0.18$ Nm, $p = 0.11$; ML training: $1.82 \pm 0.18$ Nm vs $1.93 \pm 0.23$ Nm, $p = 0.64$). Similar to the single direction baseline balance trials, comparison of pre vs. post behavior in dual axis control baseline trials (i.e., AP-baseline/ML-baseline) revealed no differences in angular velocity variance and torque SD in both the AP and ML directions (all $p > 0.10$). These results suggest that participants can rapidly transition from sustained periods of balancing with long-imposed delays to baseline standing without imposed delays.

To further assess the possibility of aftereffects from learning to balance with delays, we also evaluated the frequency characteristics of angular velocity and ankle torque signals in baseline trials before and after delay training (Figs. 4c, 5c). Comparison of pre- and post-baseline trials demonstrated that, unlike the equivalent responses during training, the frequency distributions changed in the direction that was trained. Specifically, power at frequencies from ~0.4–1.2 Hz decreased and a peak in power emerged at ~1.4 Hz for both angular velocity and ankle torque signals. In contrast, in the direction that was not trained, auto spectra were similar in pre- and post-baseline trials. Importantly, similar changes in the frequencies of control were also observed when the participants balanced in both

directions during baseline post-trials (i.e., AP-baseline/ML-baseline conditions), such that the ~1.4 Hz peak only emerged in the direction that was trained (Supplementary Note 2, Supplementary Fig. 2). Taken together, these results imply that training to balance with an imposed delay in a single direction result in small aftereffects that are isolated to the trained direction.

**Verbal reports.** In a debrief session conducted after the experiment was finished, we examined how participants perceived their change in balance control. All participants perceived a change in balance control during the delayed conditions and attributed this perception to the increased postural oscillations and the difficulty remaining within the limits (particularly before or at early stages of training). Only 2 out of the 24 training participants correctly guessed that the simulation was delayed when they were asked to describe how self-balancing control changed in the experimental conditions. Most participants (18 out of 24) instead described the simulation as being more sensitive, exaggerated, or amplified relative to their actions (i.e., describing a gain increase). The remaining participants stated they felt more unstable but were not sure how the simulation had changed. Finally, when asked how they improved their balance performance throughout training, participants (16 of 24) commonly responded that they learned to calm their control and/or make smaller motor actions with their legs and feet.

**Fig. 6 | Standing balance behavior from Experiment 2. a** Bird's eye view of whole-body angular position (°) traces of a representative participant balancing simultaneously in both AP (vertical axis) and ML (horizontal axis) directions for 20 s. These traces depict AP-baseline/ML-baseline (black), AP-delay/ML-baseline (green) and AP-baseline/ML-delay (purple) trials. During delay trials, the robotic simulator operated with a 200 ms delay. **b** Angular velocity variance highlighting the effect of an AP-delay on ML standing motion (left panel: AP-baseline/ML-baseline vs AP-delay/ML-baseline) and the effect of a ML-delay on AP standing motion (right panel: AP-baseline/ML-baseline vs AP-baseline/ML-delay). Small-filled circles represent single participants and large circles with error bars represent group averages ($n = 20$). When balancing freely in both AP and ML directions, an imposed delay in one direction increased whole-body motion variability by ~8–11× in the orthogonal (no delay) direction. Error bars represent SEMs. *** indicates $p < 0.001$.

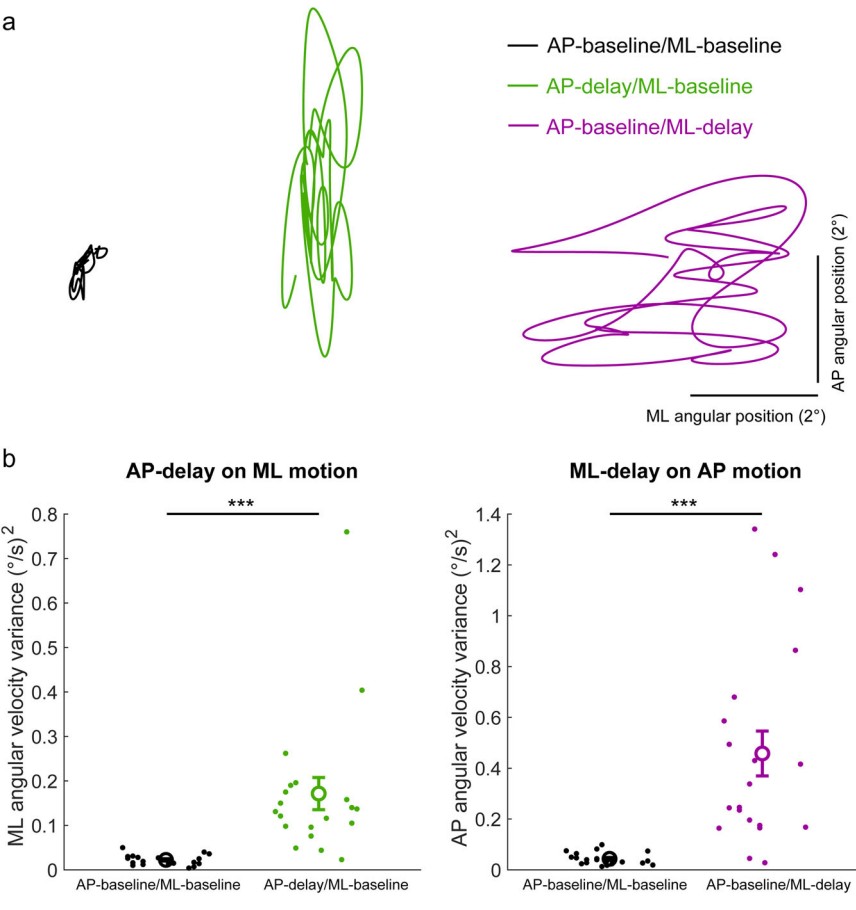

Interestingly, some participants (5 of 24) stated that they were not certain what they did despite demonstrating clear improvements in balance performance with the imposed delays. The remaining participants (3 of 24) stated their improvement simply arose naturally with practice. The verbal reports of learning to calm or reduce their motor actions suggest that participants felt they reduced their motor gain to improve balance performance with the imposed delay.

**Control experiment**. Finally, we performed a control experiment to determine whether exposure to balancing on the robot (and not training with delays) was responsible for differences between pre- vs post-learning. Here, a separate group of participants ($n = 10$) performed a near-identical experiment as above (pre- and post-learning delay trials with a 350 ms delay), with the training session being replaced by 60 min of baseline standing on the robot in the AP direction (i.e., 4 ms delay training). Our analyses showed no changes in balance behavior during pre- and post-learning delay trials (Supplementary Note 3, Supplementary Fig. 3). This excludes the possibility that the experience in pre-learning delay trials or the extended time balancing in the robotic system was responsible for the balance improvements observed in the post-learning trials for the delay training groups. Similar to the training groups, all participants reported that balancing with delays felt different from the baseline conditions. One of the ten participants correctly reported that the simulation was delayed in the delay trials while all other participants described the simulation as being more sensitive to their actions.

### Experiment 2: Delays imposed in one direction reveal potential biomechanical interactions influencing multidirectional standing behavior

The results from Experiment 1 demonstrate that training with sensorimotor delays in one direction of standing balance transfers to both the trained and untrained orthogonal directions. This transfer of balance improvements could arise from the inherent biomechanical interactions between the two directions of standing. For instance, although we restrained training to a single direction, participants may have adjusted motor commands to muscles that contribute to both directions of balance (e.g., triceps surae muscles)[39,41,42]. In Experiment 2, we assessed whether these biomechanical interactions influenced whole-body standing behavior between the AP and ML directions. A new group of participants ($n = 20$) were instructed to maintain standing balance on the robotic balance simulator for short periods (20 s) while a 200 ms delay was imposed between their self-generated ankle torques and resulting whole-body motion. This delay was chosen to maximize the amount of time participants balanced within the limits while disrupting steady stance (see Methods). Here, participants balanced in eight separate conditions, where motion was free in only one or both directions of standing (i.e., AP and/or ML), and delays were imposed in one, both, or neither of the directions.

First, we examined whether the destabilizing effects of an imposed delay in one direction influenced the orthogonal direction of baseline standing balance. When participants stood freely in both AP and ML directions with a 200 ms delay imposed in only one direction, standing balance became more variable in both directions (see Fig. 6 for representative and group data). Angular velocity variance in the direction orthogonal to the imposed delay increased by ~8–11× compared to baseline standing without delays (AP delay effect on ML velocity: 0.17 vs 0.02 $(°/s)^2$; $t_{(19)} = 4.1$, $p < 0.001$; ML delay effect on AP velocity: 0.46 vs 0.04 $(°/s)^2$; $t_{(19)} = 4.78$, $p < 0.001$). This reveals the contribution of biomechanical interactions between standing directions and demonstrates that an imposed delay in one axis destabilizes both directions of motion. The influence of these biomechanical interactions on multidirectional standing balance suggests that they may be, at least partially, responsible for the transfer of training benefits observed in Experiment 1.

To further assess how biomechanical interactions influenced standing behavior between the AP and ML directions, particularly in the context of balancing with imposed delays, we compared angular velocity variance between conditions where motion was free in only one or both directions (Supplementary Note 4, Supplementary Fig. 4, Supplementary Table 1). We first established that balancing behavior in a single direction of baseline balance was unchanged when the orthogonal direction was fixed or free (e.g., AP baseline/ML fixed vs AP baseline/ML baseline). Our analysis then showed that the disruptive effect of a delay in one direction does not depend on being fixed or free in the orthogonal direction (e.g., AP-delay/ML fixed vs AP-delay/ML baseline). We also found that the destabilizing effects of a delay did not differ when imposing delays in a single direction or simultaneously in both directions (e.g., AP-delay/ML baseline vs AP-delay/ML-delay). Collectively, these comparisons suggest that the influence of AP-ML biomechanical interactions on multidirectional standing behavior is only evident when assessing the effect of an imposed delay on the orthogonal (no delay) direction (Fig. 6). Finally, in the debrief session, we observed that all participants were aware that the delay trials were different than the baseline trials and only 1 of the 20 participants perceived the simulation as being delayed in these trials. The remaining participants described the simulation as being more sensitive to their actions ($n = 13$) or stated they were not certain ($n = 6$) how it had changed.

## Experiment 3: Generalization of balance learning between hand and leg effectors

The results of Experiment 2 support the possibility that the transfer of learning between the AP and ML directions observed in Experiment 1 could depend on biomechanical interactions underpinning the bipedal control of balance. In Experiment 3, we tested whether balance improvements gained from training generalize in the absence of biomechanical interactions by examining the transfer of training between biomechanically independent muscle effectors (i.e., hand and leg muscles). Here, we used the robot to perform balance trials where only hand-generated forces could control whole-body motion. A new group of participants ($n = 12$) trained to balance with a 300 ms imposed delay over six 3-min trials (to limit fatigue, see "Methods") when controlling their AP whole-body motion with their hand muscles (see Fig. 1 and "Methods"). This delay was chosen because it destabilizes balance in both hand-control and leg-control conditions and balance control can be improved through training in the hand-control condition (see "Methods"). After a three-minute familiarization trial without delays, eleven participants could perform a two-minute hand control baseline trial without exceeding the simulated AP balance limits (one participant briefly crossed the limits). Participants were then trained in the hand-control condition with a 300 ms delay and after training we examined whether any learning transferred to equivalent delay conditions while balancing with the leg muscles.

In all pre-learning delay conditions (legs and hand control, sample participants in Fig. 7a), participants oscillated with large angular velocity variance (legs AP-delay: $14.3 \pm 1.5$ (°/s)$^2$; legs ML-delay: $10.3 \pm 1.5$ (°/s)$^2$; hand AP-delay: $16.1 \pm 2.3$ (°/s)$^2$) and had difficulty remaining within the balancing limits for the entire trial (legs AP-delay: $68 \pm 2$%; legs ML-delay: $72 \pm 1$%; hand AP-delay: $77 \pm 2$%). Through training in the delayed hand-control condition, participants progressively improved their balance performance (Fig. 7c), reducing the variability of their angular velocity and increasing the percentage of time they were balancing within the virtual limits. First-order-exponential fits to angular velocity variance and percent time within the limits estimated mean time constants of $18.9 \pm 8.7$ min and $13.0 \pm 2.9$ min, respectively. While there were evident balance improvements in the Hand AP-delay condition from training, no participant could maintain balance for a continuous 60-second period without reaching the virtual limits.

Although participants only trained to balance their whole body with delays using their hand muscles, balance behavior was improved in post-learning trials for all hand and leg conditions (Fig. 7b). Pre-post pairwise comparisons of the hand balancing conditions revealed that participants reduced their AP angular velocity variance by ~63% ($16.1 \pm 2.3$ (°/s)$^2$ to $5.9 \pm 0.7$ (°/s)$^2$; $t_{(11)} = 5.36$, $p < 0.001$) and increased their percent time within AP limits from 77% ± 2% to 87% ± 2% ($t_{(11)} = -7.32$, $p < 0.001$). Similarly, when balancing with the legs in the AP direction, participants reduced their AP angular velocity variance by ~36% ($14.3 \pm 1.5$ (°/s)$^2$ to $9.1 \pm 1.2$ (°/s)$^2$; $t_{(11)} = 3.02$, $p < 0.05$) and increased their percent time within limits from 68 ± 2% to 81 ± 2% ($t_{(11)} = -6.13$, $p < 0.001$). Equivalent changes were also observed when balancing with the legs in the ML direction, as participants reduced their ML angular velocity variance by ~42% ($10.3 \pm 1.5$ (°/s)$^2$ to $6.0 \pm 0.9$ (°/s)$^2$; $t_{(11)} = 4.46$, $p < 0.01$) and increased their percent time within ML limits from 72% ± 1% to 82% ± 2% ($t_{(11)} = -6.94$, $p < 0.001$). The post-experimental debrief session revealed that similar to Experiments 1 and 2, participants noticed differences between delay and baseline trials. Two of twelve participants correctly stated the manipulation was a delay, while nine indicated a more sensitive control and one was not sure how the simulation had changed. All participants reported they improved control by calming or decreasing their motor actions. Overall, these results demonstrate that learning to balance the whole body with delays using hand muscles can be generalized to untrained control of balance with delays using leg muscles. Importantly, this confirms that the brain can generalize learned control policies of balancing with sensorimotor delays across biomechanically independent muscle effectors.

## Discussion

The aim of this study was to determine whether healthy adults can generalize learning to stand with imposed sensorimotor delays and establish the mechanisms responsible for any observed transfer of learning. Our results showed that regardless of the balance condition trained (i.e., legs anteroposterior, legs mediolateral, hand anteroposterior), participants learned to balance upright with long sensorimotor delays and transferred improvements in balance to untrained conditions. Balance improvement was achieved by participants learning to reduce the variability of their motor actions. Importantly, this transfer also occurred between muscle groups that were biomechanically independent for upright balance control (i.e., hand and leg muscles). When returning to normal standing conditions after training, we observed small aftereffects (frequency distribution of whole-body angular velocity and ankle torque) that were confined to the trained direction but did not influence the overall variability of body motion or motor actions. This suggests that the balance system can rapidly transition between learned sensorimotor associations with limited effects on the ongoing control of balance. Taken together, our study revealed that humans utilize prior experience acquired from balancing with imposed sensorimotor delays to control upright posture across distinct balance conditions. These findings have implications for developing robot-assisted interventions that can help individuals overcome balance impairments due to sensorimotor delays and improve everyday postural function.

After participants were trained to stand in the robotic simulator with an imposed delay – while constrained to move to a single direction (Experiment 1) – balance behavior improved in both the trained and untrained (orthogonal) directions. While these improvements were greatest in the trained direction, the transfer of this motor learning to the orthogonal direction occurred irrespective of which direction participants trained in. Our control experiment confirmed that this improvement in balance performance was due to prolonged exposure to the delay because participants who did not train with delays, but balanced on the robot for an equivalent duration, showed no improvement in balancing with delays. From these data, we rationalized that the observed transfer of balance improvements across directions could arise from biomechanical interactions across directions, and/or a neural-based generalization of learned control policies. Importantly, these two mechanisms are not mutually exclusive and could both contribute to the observed transfer.

Biomechanical interactions across directions may promote transfer because the nervous system has already adjusted the control of certain muscles contributing to the multidirectional control of standing balance. For instance, ankle muscles (e.g., triceps surae) contributing to an upright stance can produce active and passive forces that influence joint torques in

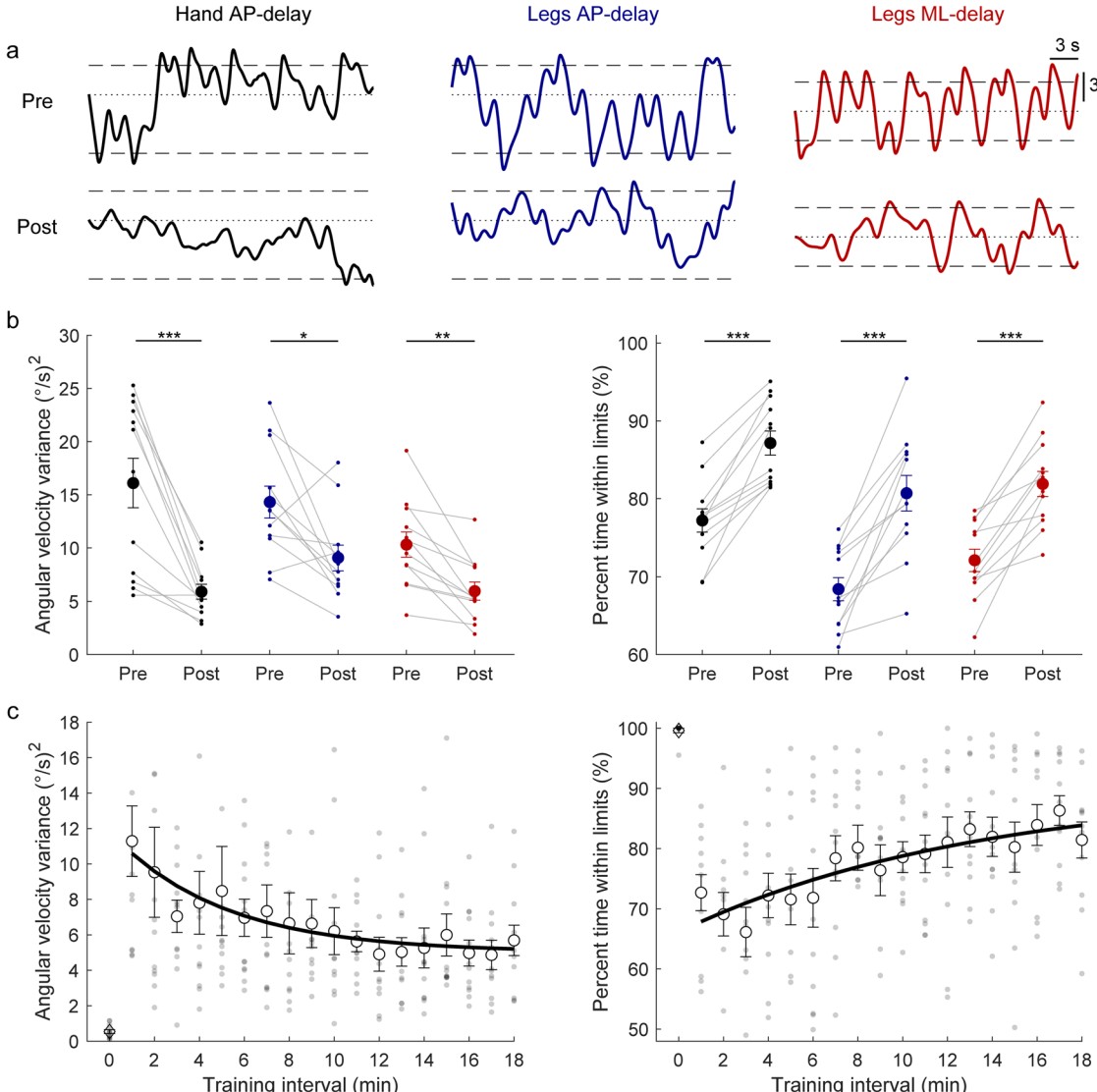

**Fig. 7 | Whole-body balance behavior from Experiment 3. a** Whole-body angular position (°) of a representative participant balancing in the hand AP-delay, legs AP-delay, and legs ML-delay trials. During delayed balance trials, the robotic simulator operated with a 300 ms delay while motion was driven either by hand-generated forces or ankle-generated torques. Whole-body motion was restricted such that motion occurred only in the AP or ML direction. Dashed lines represent the virtual position limits for AP (6° anterior, 3° posterior) and ML (3° left, 3° right) directions. Dotted lines represent the 0° position for all conditions. **b** Angular velocity variance and percent time within the limits in the pre- and post-learning trials. Thin lines connecting small filled circles are individual participants and larger filled circles are group averages ($n = 12$) with accompanying SEM error bars. Despite only training in the hand AP-delay condition, balance improvements were observed in both angular velocity variance and percent time within the limits for the legs AP-delay and legs ML-delay conditions. For all panels, data in black represent hand AP-delay trials, data in blue represents legs AP-delay trials, and data in red represents legs ML-delay trials. * indicates $p < 0.05$, ** indicates $p < 0.01$ and *** indicates $p < 0.001$. **c** Angular velocity variance and percent time within the limits for individual participants and the group average estimated over 1-minute intervals during the hand-delay training trials ($n = 12$). The solid lines show the fitting of average angular velocity variance and average percent time within the balance limits to a first-order exponential function using a least-square method: $f(x) = a * \exp\left(\frac{-x}{b}\right) + c$. Averages for one-minute balancing at baseline (4 ms delays) are represented by open diamonds.

multiple directions[39,41–43]. Indeed, results from Experiment 2 provided evidence that these biomechanical interactions may contribute to transfer of training improvements because delays imposed in a single direction increased standing movement variability in both the delayed and orthogonal directions. Crucially, Experiment 3 showed that biomechanical interactions were not required for the transfer of balance training improvements: training to balance with imposed delays in the hand-control condition transferred to the leg-control conditions in both directions of balance. Therefore, we conclude that the brain can generalize learned control policies of balancing with long sensorimotor delays across orthogonal space and biomechanically independent motor effectors.

How the nervous system facilitates this generalization may relate to what contextual features overlap across tasks[29,30,44,45]. While the primary task

to remain balanced against gravity was shared across all conditions in our experiments, contextual elements such as the biomechanics of balance control, sensory feedback, and muscles actuating body motion were different. We hypothesized that these factors may limit or prevent generalization of learning (i.e., null hypothesis). Our results, however, show that participants transferred learning regardless of these differences, supporting the possibility that the common task of upright balance may have allowed the brain to develop a general control policy that transfers between conditions (i.e., alternative hypothesis). These results align with and expand on our previous observations that training to balance in the anteroposterior direction with an imposed 400 ms delay generalizes to improve balance across other delays[21]. Collectively, our findings of generalized learning suggest that the human brain estimates the source of the distorted whole-body balance relationships

and broadly updates its control to accommodate imposed delays across directions of movement and muscles contributing to balance.

Given that sensorimotor delays vary throughout the nervous system[1,3,4] and across an individual's life[3,7,46], it is vital that the brain learns to accommodate unexpected temporal sensorimotor relationships and generalizes their effects to similar or other motor tasks. Many forms of sensorimotor learning, however, have been found not to generalize well to untrained conditions. For example, adaptation of reaching movements in force fields or with altered visual feedback leads to limited or absent transfer in other movement directions[47–51]. Similarly, adapted locomotor patterns do not transfer between forward and backward walking[52,53], and the adaptation of balance responses to perturbations while standing is constrained only to the direction practiced[38,54]. More specific to the current study, when adapting to experimentally-imposed delays, improvements in visually-guided upper limb movements (i.e., intercepting a moving target) are not generalized to other directions or tasks (i.e., manual tracking)[29,55]. This is with the exception of continuous visuomotor tasks, such as video game pong or driving, where partial transfer occurs to variations of the same task[56] or to discrete reaches to targets[57,58]. Perhaps the demand to maintain accurate motor control during continuous tasks, like standing balance, necessitates the rapid learning and generalization of skills under these conditions. The question remains what changes in sensorimotor control are learned to balance with added delays across different contexts?

One mechanism proposed to stabilize control with long delays is to reduce the magnitude of the motor action needed to respond to a deviation from the desired position (i.e., the controller gain). Indeed, computational models of human standing predict that the balance system decreases feedback control gains (i.e., proportional and/or derivative gains) to accommodate increased neural delays[17,20,59]. Our results showed partial support for this change in control. First, participants progressively reduced the variability of their ankle-produced torques. Second, in the debrief session, participants often noted that they were able to improve their balance performance by making calmer and smaller motor actions to control ongoing balance oscillations, changes that should accompany reduced control gains. Adjusting control gains for balance may also facilitate generalization, as it would be suitable for controlling balance with delays in other contexts (e.g., hand vs. leg control). A reduction in sensorimotor gain may further explain why transitioning from delayed-control to baseline balance did not result in obvious destabilizing behavior because the gain levels acquired for balancing with long delays will be suitable for balancing without delays[20,59]. We note, however, that our analysis of torque and whole-body motion are not direct quantifications of feedback gains of the closed-loop balance controller. Future studies using independent perturbations (mechanical and/or sensory)[60–62] may be able to assess the open-loop transfer function between whole-body movement (i.e., sensory feedback) and muscle activity (i.e., motor command) to directly assess changes in the neural controller. Similar approaches may also be used to test alternative hypotheses that the nervous system increases ankle joint stiffness[59,63] and/or uses intermittent control policies[64–66] to accommodate increased sensorimotor delays in balance control.

Our results also revealed changes in the frequency distribution of torque and whole-body angular velocity during baseline balance after participants had trained with the delay (see Figs. 4, 5 and Supplementary Fig. 2). Specifically, participants generated oscillatory torques at ~1.4 Hz (i.e., an ~700 ms period), ensuring that the torque to whole-body acceleration relationship was inverted relative to normal balance when in the presence of the 350 ms delay (i.e., the half period of 1.4 Hz). Similar shifts in the frequency distribution also occur during arm reaching when adapting to visually induced delays[57]. Notably, this frequency-specific strategy is different from our original expectation (see Methods) that participants would adjust the timing relationship between their motor commands and whole-body motion. Using an LQR model of balance control that predicts the disruptive effects of the delay (see Methods), we showed that the timing between torque and acceleration signals cannot reveal adaptations to the delay because the imposed robotic delay always ensured whole-body motion

occurred later than the control torque. Therefore, the production of a 1.4 Hz torque signal may be an adaptation by the nervous system to account for the influence of the delay in a compensatory manner. Despite the frequency changes, its influence on observed balance is relatively minor (i.e., small aftereffect), remaining undetected in the auto spectra of the training trials and causing no discernible changes in the overall variability of baseline pre- and post-trials. Exploring the impact of training with different delays on baseline balance may reveal delay-specific changes in control frequencies similar to those observed here.

Aftereffects that emerge after a period of motor learning are often explained to occur through implicit mechanisms of sensorimotor adaptation[27,30,67,68], whereby the motor output is recalibrated using errors between expected and actual sensory feedback. For example, implicit adaptations are observed in the vestibular control of standing balance, where vestibular-evoked responses are rapidly (dis)engaged and spatially transformed outside of conscious awareness[21,22,24,69,70]. Thus, the aforementioned changes in frequency distributions may indicate implicit alterations in the balance controller. On the other hand, the observed aftereffects from learning were small and did not influence overall variability in balance behavior. Therefore, the adaptation to balancing with delays may align better with explicit (i.e., cognitive) learning, where goal attainment (i.e., remaining upright) is dominant and small or absent aftereffects are observed[30,71–74]. The verbal reports from the debrief sessions further imply that participants were aware of changes in their overall whole-body movements and balancing actions, though they rarely perceived them as delayed control. In addition, the observed generalization of learning supports the involvement of cognitive mechanisms because the transfer of learning is more pronounced when explicit learning mechanisms are involved[30,72,75–78]. Presumably, the net changes in balance control in response to long sensorimotor delays are driven by a combination of implicit (automatic, absent of cognitive processes) and explicit (conscious awareness and cognitive strategies) mechanisms. Although our study was unable to isolate these contributions, future experiments designed for this purpose could contrast learning and generalization results in participants who train with explicit instructions to those that train without.

Our findings have important implications for the training and rehabilitation of clinical populations with balance impairments. During aging and certain diseases (e.g., diabetic neuropathy or multiple sclerosis), increasing sensorimotor delays may compromise an individual's balance control and lead to falls[5,6,43]. We previously demonstrated that humans can be trained to accommodate long-imposed delays (400 ms) in the control of standing balance and this ability is retained when participants are tested again three months later[21]. Furthermore, we have shown that older adults (>65 years) can also learn to maintain standing balance with long delays[79]. Importantly, for rehabilitation to be effective, the evoked learning needs to be generalizable because humans perform movements across a wide range of circumstances[80,81]. Our present findings demonstrate that humans generalize learning to balance with delays across a variety of contexts (i.e., different directions and muscle effectors). These results make it possible to envision future robot-assisted training therapies that go beyond current methods of repeatedly exposing participants to physical perturbations[82,83], and instead allow the nervous system to explore distinct environments and adjust to the related instabilities. Furthermore, our results show this approach to be more effective than postural disturbances at transferring learning to different balance directions[38]. We note, however, that our study is limited as it only assessed standing balance within our robotic simulator. Future studies are needed to determine whether robot-assisted sensorimotor manipulations of the balance control loop, such as those implemented here, translate to improved balance control in everyday activities.

Our data demonstrate that humans generalize learned control policies of balancing with unexpected sensorimotor delays across different directions of standing and biomechanically independent muscle effectors. While biomechanical interactions between the lower limbs influence the multidirectional control of standing balance, transfer of balance learning is not contingent on these interactions. This generalization may be achieved by the brain learning an effective motor behavior, such as a change in control gain,

to balance upright with long sensorimotor delays and applying this control policy to different variations of the balance task.

## Methods

### Ethics statement

This study was approved by the Medical Research Ethics Committee Erasmus MC and conformed to the Declaration of Helsinki with the exception of registration to a database. The experimental protocol was verbally explained to all participants and written informed consent was obtained before commencing the experiment.

### Participants

A total of 66 healthy adult participants (30 females, mean age: 25.2, SD: 3.9 years old) with no known history of neurological and/or balance deficits participated in this study (24 in Experiment 1, 10 in Experiment 1 control group, 20 in Experiment 2, 12 in Experiment 3).

### Experimental set-up

Three experiments were conducted to study whether humans can learn and generalize the ability to stand with imposed sensorimotor delays. For all experiments, participants stood on a custom-designed robotic balance simulator programmed to replicate the control of standing balance in the anteroposterior (AP) and mediolateral directions (ML) of motion (Fig. 1). This system is similar to the robot developed by Qiao et al. [84] to simulate balance control in the anteroposterior and mediolateral directions[84]. Consequently, we refer to this publication when describing the robotic system in addition to providing relevant details included in our robotic simulator.

The mechanical load of the body was simulated using a real-time motion controller (PXI-8880; National Instruments, TX, USA) running at 500 Hz. We programmed the simulation using the anthropometry of each participant, including: mass, height, center of mass height, sternal notch height, hip height, ankle height, and hip joint width. To measure the height of their center of mass, participants laid supine on a rigid board that was balanced over a round tube positioned transversally under the board. Participants shifted their body longitudinally over the board until the distribution of their mass was balanced. The distance between their medial malleolus and the tipping point was determined as the height of the center of mass (average: 0.90, SD: 0.04 m). The robotic apparatus consists of a rigid backboard frame, an ankle-tilt platform, a torso harness, and a pelvis harness, each controlled by a separate servo motor-driven linear actuator[84]. The entire system has an ~4 ms delay between a position command and the measured position change of the motors. The backboard frame is driven by a 2 kW servo motor (ECMA-J11020S4, Delta, Taiwan; maximum continuous torque: ~6170 Nm; angular resolution of ~0.0000054°) connected to a 665 mm linear actuator (Y-H1116165P09152A; Rollon, Italy). The ankle tilt platform is driven by a 400 W servo motor (ECMA-C10604SS, Delta, Taiwan; maximum continuous torque: ~234 Nm; angular resolution of ~0.000014°) connected to a 665 mm linear actuator (Y-H1116105P05442A; Rollon, Italy). Both the torso and pelvis harnesses are driven by 400 W servo motors (ECMA-C10604SS, Delta, Taiwan; maximum continuous torque: ~780 Nm (pelvis) and ~1170 Nm (torso); angular resolution of ~0.00000382° (pelvis) and ~0.00000573° (torso)) connected to 586 mm linear actuators (Y-H1116105P08362A; Rollon, Italy). Participants wore noise-cancelling headphones (WH-1000XM3 Noise Cancelling Headphones, Sony, Japan) with audio of garden sounds (water fountain with birds singing) to minimize acoustic cues of motion produced by the motors as well as other extraneous sounds.

The robotic simulator implemented the dynamics of a single-link inverted pendulum to control whole-body motion in the AP and ML directions. Specifically, the simulator used the following continuous transfer function that was converted to a discrete-time equivalent state-space model for real-time implementation using a zero-order hold method:

$$I\ddot{\theta} - 0.971mgL\theta = T$$
$$\frac{\theta}{T} = \frac{1}{Is^2 - 0.971mgL} \tag{1}$$

where $\theta$ is the angular position of the body center of mass around the axis of rotation from vertical. In the AP direction, $\theta$ is positive for a backward-leaning center of mass position and in the ML direction, it is positive for a left-leaning center of mass position. $T$ is the ankle torque applied to the body; it is positive in the AP direction for a plantar-flexor torque and in the ML direction for a greater load on the right leg. $m$ is the participant's total mass, $L$ is the distance from the ankle joint to the body center of mass, $g$ is the gravitational acceleration (9.81 m/s$^2$) and $I$ is mass moment of inertia of the body measured about the ankles ($0.971mL^2$). The body mass above the ankles can be approximated by removing the estimated weight of the feet from the participant's total body weight such that the effective mass is estimated as $0.971m$. While a participant controls the robot, their body center of mass rotates about the ankles as in normal standing. Under these conditions, the passive stiffness and viscous damping properties of connective tissue are provided by the rotation of the participant's ankle.

In the ML direction, the simulator uses the same inverted pendulum mechanics but distributes the whole-body angular position to the pelvis and torso harnesses to control lateral translation at the height of the pelvis and shoulders according to mechanical models[31,33,84]. The equations for motion in ML are influenced by stance width, such that when the feet are next to each other the motion of the torso and pelvis are closer to the movement of an inverted pendulum. Here, we were interested in having similar movement kinematics for both AP and ML directions and therefore participants maintained a narrow stance width of ~8 cm between ankle joint centers in all experiments.

When the simulation is engaged, participants control their whole-body movements based on the torques applied to the force plate (Figs. 1, 8). Forces and torques generated by the participant are transformed to the midpoint of the ankle joints to provide net ankle torques in the AP (Tx) and ML(Ty) directions. These torques are used as inputs into the inverted pendulum state-space models of AP and ML standing balance operating in parallel[84] to drive center of mass (CoM) motion. The resolved AP CoM angle dictates the angle of the backboard and footplate, and the ML CoM angle is further decomposed into linear movements of the pelvis and torso. The ML CoM decomposition predicts the positions of the pelvis and torso for various stance widths under the assumption that the pelvis is perpendicular to the torso (see ML decomposition in Qiao et al. [84]). As these four motors can be independently engaged/disengaged, this facilitates different motion combinations such as normal balance, balancing only in AP or ML directions, and ankle sway referencing.

The robotic system functions through a two-layer architecture (Fig. 8) similar to Qiao et al. [84]. The higher layer involves an embedded controller (PXI-8880; National Instruments, TX, USA) with a data acquisition module that digitizes the forces and moments sensed by the force plate and runs the robotic simulation at 500 Hz. Target encoder counts from the higher layer are sent to the FPGA module (NI PXI-7846R) at the lower layer and the FPGA communicates directly with the four motors. Based on the current and target encoder count, the FPGA sends a target torque command as an analog signal (voltage) to each motor's servo drive. Running at 40 MHz, the FPGA counts the encoder pulses of all motors (i.e., measured angular position) and executes a four-axis torque feedback proportional-integral-derivative controller to actuate motions at 2000 Hz.

Throughout the experiments, all signals were recorded through a data acquisition board (PXI-6289; National Instruments, Austin, TX, USA) and digitized at 500 Hz. While secured to the robot through the harnesses, participants stood on a force plate (AMTI BP400 × 600; Watertown, MA, USA) which measured and amplified (×4000) ground reaction forces and torques. The force plate was securely mounted on top of the ankle-tilt platform which was kept horizontal (replicating standing on level ground) in balance trials controlled by the leg muscles. When participants controlled the robot in the AP direction with forces generated with their intrinsic hand muscles (see Experiment 3), they modulated force by abducting/adducting their right index finger against a fixed load cell (BOSCHE; S-Type, Damme, Germany). This load cell measured forces in a single axis and a strain gauge signal conditioner was used to amplify the signal for data acquisition

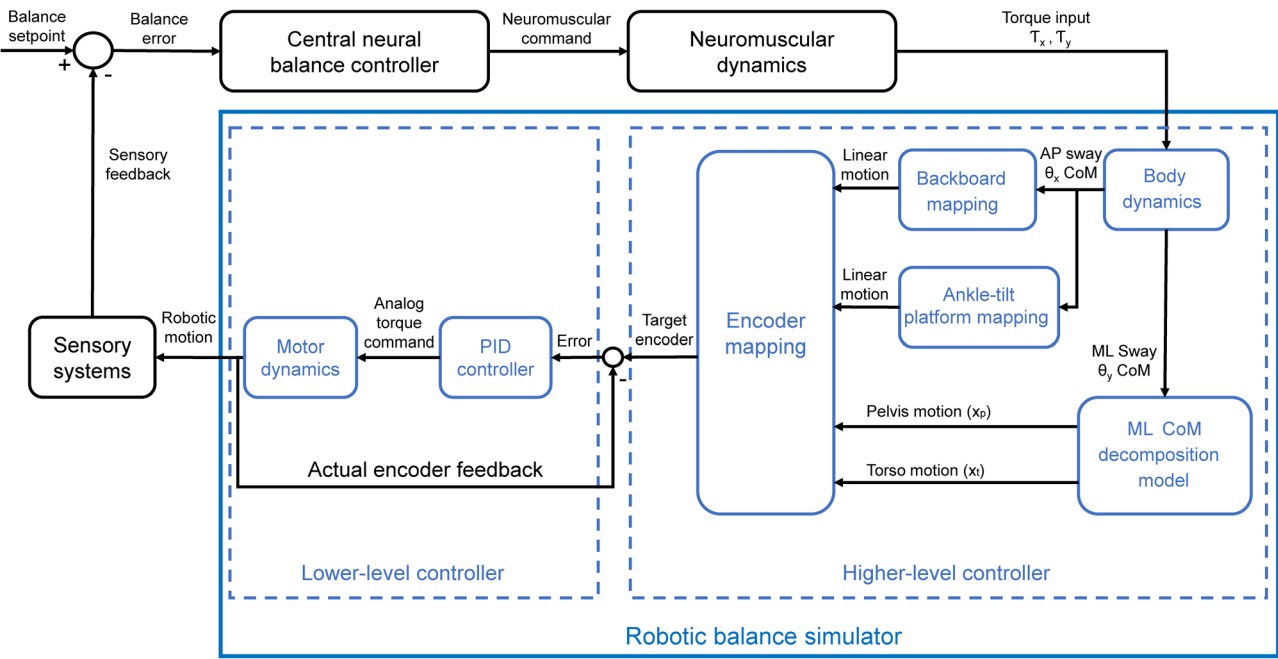

**Fig. 8 | Robotic balance simulation control diagram.** Net ankle torques in the anteroposterior ($T_x$) and mediolateral ($T_y$) directions are used as inputs for independent state-space models of AP and ML body dynamics (see Eq. 1) that solve for whole-body CoM angles related to standing. CoM position in the AP ($\theta_x$) direction is mapped onto the linear position of the motors controlling the backboard and ankle-tilt platform, enabling different motion combinations (i.e., normal balance and ankle sway referenced). CoM position in the ML ($\theta_y$) direction is decomposed into linear motion of the pelvis ($x_p$) and torso ($x_t$) under the assumption that the torso is perpendicular to the pelvis[17]. The control diagram is adapted from Qiao et al.[84], where equations describing the ML-motor decompositions can be also found.

(FUTEK IAA100; Irvine, CA, USA). The load cell was fixed to the proximal interphalangeal joint of the extended right index finger to measure the force generated when isometrically contracting the intrinsic hand muscles. The output of the load cell was scaled for each participant so that generating 20% of the maximal voluntary adduction force maintained an anterior leaning posture of 2.0°. This was selected as pilot experiments indicated this scaling of force allowed participants to perform the hand-control trials and generate enough force to bring the whole body upright if they exceeded the virtual limits. Maximal voluntary contraction force was computed by performing three trials where participants attempted to generate maximal force against the load cell for ~3 s. Peak force was extracted from each trial and the average peak force across the three trials was used as the maximal voluntary contraction. During hand control balance trials, the ankle tilt platform rotated in the same direction as a whole-body motion to maintain a constant ankle angle throughout the trials – an approach known as ankle sway-referencing[60,85]. This was implemented to remove the sensory conflict that arose from ankle somatosensory feedback not being coupled to ankle-generated torques to control balance. During normal anteroposterior standing, musculoskeletal tissues around the ankle are deformed leading to passive viscoelastic forces contributing to stabilizing upright stance[14,15,86,87]. In hand control conditions, however, forces and torques generated at the feet have no influence on whole-body motion, and normal passive contributions from the deformation of ankle tissues are absent. Therefore, the normal ankle torque contributions of passive stiffness[88] and damping[14] arising from the ankles were simulated for each participant using properties derived from anteroposterior standing. Specifically, an additional damping term was included in the inverted pendulum transfer function[89], while the effect of passive stiffness incorporated both short- and long-range properties of muscle stretch[88].

Seatbelts placed around the shoulders/chest and the waist secured the participant to the torso and pelvis harnesses and prevented them from falling forward without supporting the load of the body acting through the feet. The harnesses are lined with medium-density foam and a layer of foam was placed between the seatbelts and the participant at the chest and waist. In the AP direction, the backboard frame rotated the body about an axis

collinear with the axes of the ankle joints. Angular motion of the body in this direction was restricted using software limits to a maximum angular position of 6° anterior and 3° posterior to ensure that participants could generate sufficient torque to balance the system across the range of motion[21,69,89]. In the ML direction, the pelvis and shoulder harnesses controlled the motion of the body and were supported by separate gas springs to avoid imposing any vertical load on the participant. This also allowed for the pelvis and shoulder harnesses to be aligned at the level of the greater trochanter and sternal notch. Angular motion of the body in this direction was restricted using software limits on the inverted pendulum model of 3° left and 3° right from vertical. These limits were chosen to ensure that while participants stood at the narrow stance width (~8 cm), they did not shift their feet or take a step when reaching the limits of position. For both AP and ML control, when participants exceeded the software position limits, the program gradually increased the simulated stiffness such that participants could not rotate further in that direction regardless of the torques they produced at the ankle. This was performed by linearly increasing a passive supportive torque to a threshold equivalent to the participant's body load over a range of 1° beyond the simulated balance limits, providing a passive support of the body at that angle. Active torques applied by the participants in the opposite direction enabled them to get out of the limits. An additional damping term over the range of 1° was implemented to ensure a smooth attenuation of motion.

In the present experiments, we imposed delays between the participant-generated ankle torque or finger force (i.e., motor command) and the resulting whole-body motion (i.e., sensory feedback). Delays were imposed by buffering participant-generated torque or force recordings such that the signals driving motor position commands (thus whole-body motion) could be delivered based on the torque or force participants generated up to 500 ms in the past. It is worth noting that the natural sensorimotor delays within human standing balance control are ~100–160 ms[19,85,90]. Therefore, these natural delays need to be added to the imposed delays to estimate the overall standing balance control delays. Throughout this study, we refer to the delays added through robotic simulator (baseline (4 ms)-350 ms), but the total sensorimotor delays for the standing balance task are ~100–160 ms larger. All participants were naïve to the delay protocols and were simply told that: in

some trials, the robotic control will be changed, such that your body movement may seem unexpected or abnormal, and standing balance may become more difficult. However, during these conditions, you will still be in control of your body movement. In all experiments, participants were instructed to stand upright normally at their preferred standing angle (typically ~0–2 ° anterior and ~0 ° in the mediolateral direction). In trials with delays ≥300 ms, participants had difficulty maintaining a stable upright posture and would often exceed the simulated balancing limits (i.e., 6 ° anterior or 3 ° posterior; 3 ° left or 3 ° right). Participants were instructed to always get out of the limits and continue to attempt to balance upright. After a trial was completed, the robot was returned to a neutral position (0 ° AP, 0 ° ML) at a fixed velocity (0.5 °/s) in preparation for the next trial.

## Familiarization

For all experiments, an introductory balance session was first completed to familiarize participants with the control of the robot. Instructions were given on the nature of movement control; i.e., similar to standing, applying torque to the support surface (force plate) will control the motion of the upright body (via backboard frame, torso, and pelvis harnesses). Participants were familiarized with the baseline (i.e., 4 ms delay) control of the robot in three conditions: free-standing control (simultaneous AP- and ML-control), AP-control, and ML-control. In the AP- and ML-control conditions, the motors for the orthogonal axes remained stationary. For AP motion (free standing or AP-control), a plantar-flexor torque is required to stabilize the body when standing in a forward-leaning position. Increasing the plantar-flexor torque greater than the gravitational torque will cause the body to accelerate backward. Similarly, a dorsi-flexor torque is required when standing in a backward leaning position and an increase in dorsi-flexor torque will accelerate the body forward. For ML motion (free standing or ML-control), maintaining a greater load on the left leg is required to stabilize the body when standing in a leftward leaning position. Similarly, maintaining a greater load on the right leg is required to stabilize the body when standing in a rightward leaning direction. During the familiarization session, participants were instructed to move their body in all directions and allow the robot to reach its limits (6 ° anterior, 3 ° posterior, 3 ° left, 3 ° right), which occurs if the magnitude of the generated ankle torque is not large enough to resist the toppling torque of gravity. Participants performed this familiarization period until they were accustomed to standing balance on the robot and could maintain an upright posture at these baseline conditions with ease. Participants were told that these familiarization conditions were the baseline conditions of the robot, and that while it may be more difficult to balance upright for some experimental trials, participants would always be able to control their motion by adjusting how they loaded and pushed their feet against the force plate (or pushed their finger against the hand device in Experiment 3). The entire familiarization session was completed within 10 min.

## Experimental protocol

Across the experiments, we examined several variations in balance conditions. As the robotic simulation independently controls whole-body motion along the AP and ML directions, we defined each condition by parameters set for each balance direction. Each direction can be: (1) baseline control (i.e., AP-baseline), (2) fixed, such that no motion occurred (i.e., AP-fixed), or (3) delayed, with imposed time delays (i.e., AP-delay). Finally, in Experiment 3 we further denoted whether the robot is controlled by the legs or the hand (i.e., legs AP-delay; legs ML-delay; hand AP-delay).

**Experiment 1: learning to stand with imposed delays and transfer between balance directions.** In our first experiment, we designed a training protocol to determine if learning to balance with the imposed delay in one direction benefits balance with imposed delays in the untrained orthogonal direction (i.e., transfer of sensorimotor learning). Participants (n = 24) were divided into one of two groups in which they were trained with a 350 ms imposed delay in one direction of balance

while the orthogonal direction was fixed: AP-delay/ML-fixed training (n = 12) or AP-fixed/ML-delay training (n = 12). We chose an imposed 350 ms delay (with added internal delays, net ~450–510 ms) because participants initially cannot maintain standing balance for more than a few seconds with these delays, and therefore, require training to regain balance[21]. All participants completed the same pre-learning and post-learning sessions, consisting of ten 30 s trials. Five conditions were each tested twice: AP-baseline/ML-baseline, AP-baseline/ML-fixed, AP-fixed/ML-baseline, AP-delay/ML-fixed, and AP-fixed/ML-delay (delay = 350 ms). To minimize learning within and between trials (and prior to the training session), we limited the trial duration to 30 s, providing a total of 60 s of data for each condition. Trial order was pseudorandomized such that the first trial of each condition was completed before performing any of the second trials (i.e., five conditions randomly ordered followed by the same five conditions with a new random order). Participants then performed a 60-minute training session, where they trained to balance with a 350 ms delay over six 10-minute periods. Balance control was restricted to a single direction during training such that for a given training group, all six training periods were of the same balance condition (e.g., AP-delay/ML-fixed). Immediately following the training session, participants then performed the post-learning session (same as pre-learning but with a different pseudorandom trial order). Finally, we performed a separate control experiment to determine how the pre-learning exposure to the delays and prolonged experience of balancing on the robot influenced performance in the post-learning trials. Here, a new group of participants (n = 10) performed the same experimental protocol as above, but the training session was replaced with a 60-minute session of balancing in the AP-baseline/ML-fixed condition.

**Experiment 2: multidirectional control of balance with imposed delays.** In Experiment 2, we characterized the effects of biomechanical interactions on the multidirectional control of standing balance with delays. Because of the architecture of lower limb muscles (i.e., attachment points and fiber angles) and because both feet generate ground reaction forces and torques to control upright balance, active or passive muscle tensions can influence joint torques in multiple directions of standing[39–43]. Therefore, we hypothesized that an induced delay in one direction of balance would affect whole-body oscillations in both directions. To test this, we examined participants' standing balance in both AP and ML directions when a 200 ms delay was imposed in one or both directions. We chose a 200 ms imposed delay for all conditions because it increases standing balance variability while limiting the number of times participants exceed the balance limits[21], thus ensuring participants were actively balancing for the majority of the trial (which was needed for our analysis, see Data reduction and signal analysis). Participants (n = 20) balanced the robotic simulator with body movements fixed in one direction or free in both directions. Each participant first completed three 120 s baseline trials with control in either one or both directions (AP-baseline/ML-baseline, AP-baseline/ML-fixed, AP-fixed/ML-baseline) presented in a random order. Participants then completed ten 20 s trials with imposed delays in one or both directions of balance. Delay trials were limited to 20 s in duration to minimize any learning within and between trials. Five conditions (each performed twice) were tested: AP-delay/ML-fixed, AP-delay/ML-baseline, AP-fixed/ML-delay, AP-baseline/ML-delay and AP-delay/ML-delay. The primary aim of these conditions was to determine whether imposed delays in one direction affected standing behavior in the orthogonal direction. To control for ordering and cross-over effects between trials, we used a balanced Latin square design.

**Experiment 3: transfer of learning across independent muscle effectors.** Experiment 1 demonstrated that training in one direction of standing balance benefited both the trained and untrained orthogonal directions (see "Results"). The transfer of training benefits may arise from biomechanical factors such as the interactions between directions of standing balance. In Experiment 3, we examined whether these biomechanical interactions are required for transfer of balance learning by

testing whether learning transfers between muscle effectors that are mechanically independent (i.e., hand and leg muscles). Participants (n = 12) completed balance trials where they controlled the robot either with their legs (i.e., traditional standing balance control) or with their hands. In hand control trials, the robot was programmed such that whole-body motion could only be controlled using hand-generated forces through the contraction of hand muscles. Here, participants balanced the robotic system by modulating abduction/adduction isometric force generated at the second metacarpophalangeal joint of the right hand (i.e., the right index finger; see Fig. 1).

Participants performed balance trials under baseline conditions and with a 300 ms delay. Here, we chose a 300 ms delay to minimize hand muscle fatigue throughout the experiment and ensure that participants could learn to balance their upright body after an initial period of instability. Participants first completed three 120 s baseline trials balancing with the legs in AP and ML directions and the hand in the AP direction under baseline conditions (one trial per condition, order randomized across participants). Participants then performed a pre-learning testing session consisting of six 30 s trials. Three conditions (each performed twice) were tested: (1) balancing in AP (with ML fixed) using the legs and an imposed 300 ms delay, (2) balancing in ML (with AP fixed) using the legs and an imposed 300 ms delay, and (3) balancing in AP (with ML fixed) with the hand and an imposed 300 ms delay. To minimize learning within and between trials (and prior to the training session), we limited the trial duration to 30 s. Trial order was pseudorandomized such that the first trial of each condition was completed before performing any of the second trials (i.e., three conditions randomly ordered followed by the same three conditions with a new random order). This provided a total of 60 s of data for each condition. Participants then performed an 18-minute training session, where they trained to balance their whole body in the AP direction with their hand and the imposed 300 ms delay. The training was completed over six 3-minute trials. We limited the total length of training to 18 min and limited each trial to 3 min in duration because pilot experiments demonstrated that the hand muscles fatigued after training for longer periods. Furthermore, pilot experiments demonstrated that participants showed clear balance improvement after 18 min of training. Between each training trial, participants rested their hand by removing it from the hand device. Participants were also given a 5-minute break after the 2nd and 4th training trial where they came off the robot and sat in a chair. After the training session was complete, participants performed the post-learning session, which was identical to pre-learning with a new trial order.

**All experiments: debrief session.** At the end of each experiment, we conducted a short debrief interview with the participants to obtain qualitative information about their experience balancing with the imposed delays. During set-up and testing, participants were not explicitly told that delays were imposed on the robotic simulation, and therefore, we were interested in documenting their perception of the delay when balancing and training in the imposed delay trials. First, we asked participants if they noticed a difference in control between the experimental conditions (i.e., imposed delay trials) and baseline conditions (i.e., 4 ms delay). We then asked them to describe the differences they noticed between experimental and baseline control trials (i.e., how did whole-body movement control change between conditions). Finally, we asked them to describe how they changed their behavior to improve their balance in the experimental conditions. This final question was of particular interest to participants who trained with the delays (Experiments 1 and 3) to determine whether they were consciously aware of strategies they used to improve their balance performance.

**Data processing and analysis**

**Measures of balanced behavior.** All non-statistical data processing and analyses were performed using custom-designed routines written with Matlab software (2022a version, Mathworks, Natick, MA, USA). Across experiments, we extracted whole-body angular velocity variance and

ankle torque standard deviations as measures of balance behavior. Both measures were only estimated from data in which whole-body angular position was within the virtual position limits (AP: 6 ° anterior and 3 ° posterior; ML: 3 ° left and 3 ° right) because standing with delays ≥ 200 ms can result in participants crossing the balance limits[21]. When first standing with large delays (i.e., first exposure to delays ≥300 ms), participants often only balanced within the limits for short periods (~2–5 s). Therefore, in order to extract meaningful angular velocity and ankle torque information throughout the entire trial, we extracted data in non-overlapping 2 s windows when participants remained within the simulated balance limits. Data extracted were limited to multiples of 2 s, such that if there was a 5-s segment of continuous balance, only the first two 2 s windows (i.e., first 4 s) were extracted. On a participant-by-participant basis, we then averaged angular velocity variance and torque standard deviation estimated from these 2 s windows for each participant in each experimental condition[21]. In the training trials, angular and torque measures were estimated from non-overlapping 2 s windows taken across 1-minute intervals throughout the training. They were then further averaged across all participants, providing a minute-by-minute representation of balance behavior throughout training. After demonstrating that changes in ankle torque standard deviations aligned with changes in angular velocity variance in Experiment 1 (see "Results"), we only presented angular velocity variance in Experiments 2 and 3. In Experiments 1 and 3, we also computed the percentage of time that the whole-body angular position remained within the simulated balance limits[21]. This was computed over 60-second intervals for training trials as well as the pre-learning and post-learning conditions (by combining the two 30 s trials). Across all experiments, baseline (no delay) trials were analyzed in the same manner as trials with imposed delays. Whenever we compared baseline trials to delay trials, we cut the baseline data to be of equivalent length as the delay data if the delay trials were shorter (i.e., Experiment 2: 40 s used for both baseline and delay trials).

**Statistics and reproducibility.** For all three experiments, statistical analyses were performed using SPSS software (version 23.0, IBM), and the significance level was set at 0.05. All data were checked for normality using Shapiro-Wilk tests. Group data in text, tables, and figures are presented as means with accompanying SEMs unless otherwise specified. Below we present the relevant statistical analysis for each experiment.

**Experiment 1.** For Experiment 1, all analyses were performed using measures of whole-body angular velocity and position (i.e., angular velocity variance and percentage of time within the limits) and balancing motor actions (i.e., ankle-generated torques). To determine learning rates on a participant-by-participant basis, we fitted first-order exponential functions to the velocity variance, percentage of time within the limits, and torque standard deviation obtained over the 60 minutes of training. To compare learning rates between AP and ML training groups, we performed independent sample t-tests on the time constants (time of a 63.2% improvement) extracted from the exponential function fits. We expected that training would improve balance behavior, thereby reducing angular velocity variance, increasing the percent time within the balance limits, and decreasing ankle torque variability. To test if balance improvements gained from training in AP transferred to ML and those gained from training with ML transferred to AP, we used one-tailed paired t-tests to compare balance behavior in pre- vs. post-learning for both the AP-delay and ML-delay conditions. For the control experiment, we tested whether exposure to balancing on the robot (and not training with delays) led to balance improvements (i.e., changes in angular velocity variance or ankle torque SD) in the post-learning compared to pre-learning trials by using one-tailed paired t-tests. To determine whether training performance transferred to the untrained directions, we assessed the relative improvements in balance performance between the trained and untrained balance directions. On a participant-by-participant basis, we calculated the percent improvement from pre- to

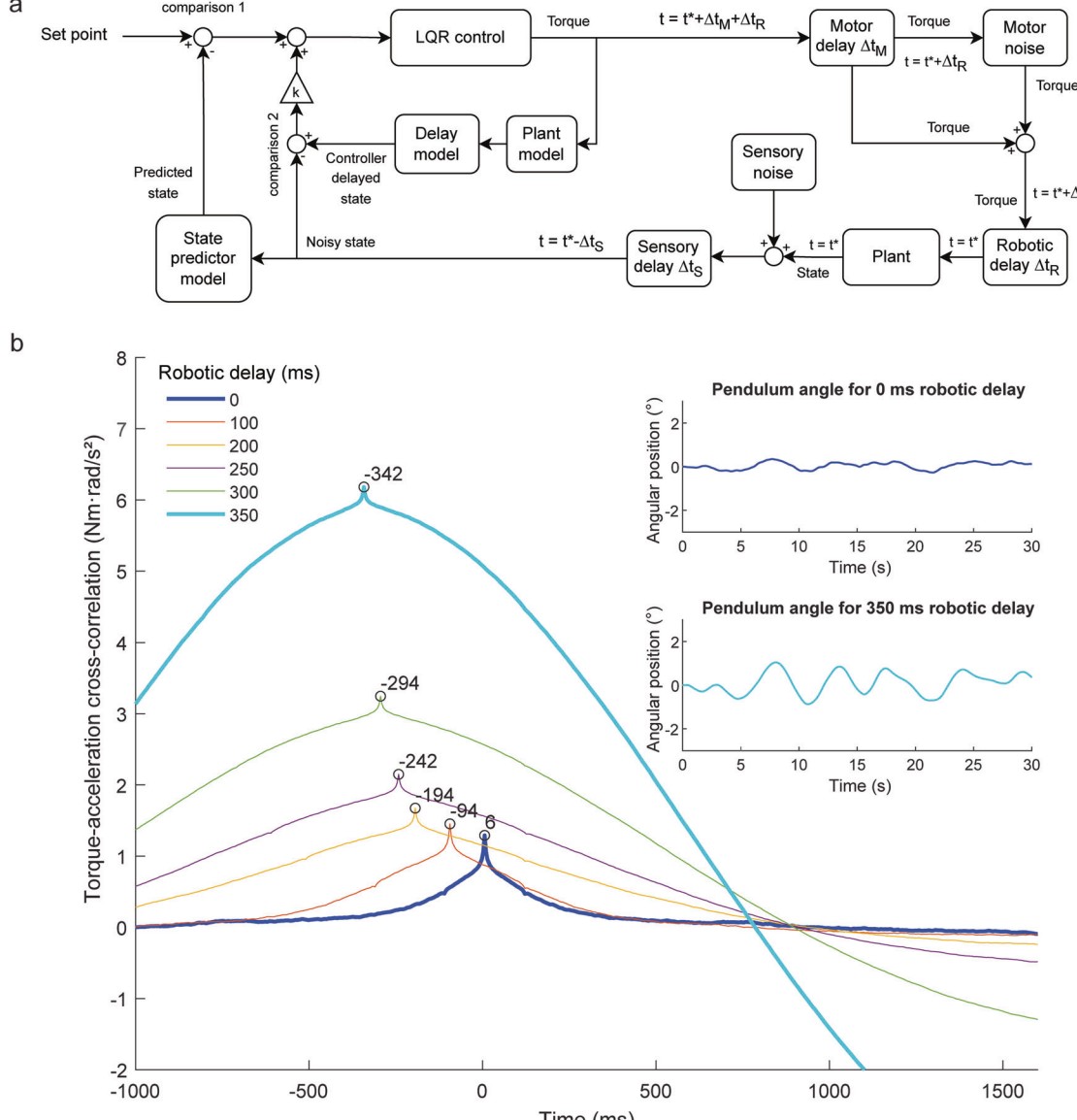

**Fig. 9 | Simulation of controlling balance with imposed delays using an LQR controller and Smith predictor. a** Control system architecture with LQR feedback loop using a Smith predictor. This control diagram illustrates the closed-loop control system architecture integrating a Linear Quadratic Regulator (LQR) for optimal state feedback. The system dynamics are captured by the plant model, with the inclusion of delay elements and noise to simulate real-world conditions. The delay for the motor system, robotic system, and sensory system are labeled $\Delta t_M$, $\Delta t_R$, and $\Delta t_S$ respectively. The torque generated by the controller is delayed and a motor noise proportional to the torque is added. Sensory noise is also introduced to the state observed from the plant. The Smith predictor is comprised of a plant model, delay model and state predictor model. Feedback from the state predictor model is compared to the set-point (comparison 1) to provide an error in the desired state. An additional error term is estimated by comparing the controller delayed state (the expected consequence of an action generated at $t = t^* + \Delta t_R + \Delta t_M$ and observed at $t = t^* - \Delta t_S$) to the noisy state (comparison 2). This error is also multiplied by a gain term (k), with a value less than 1, that is used to ensure that the set-point error is dominant. The two errors are summed as input to the LQR controller, which in turn drives the inverted pendulum motion. **b** Cross-correlation estimates between the torque applied to the inverted pendulum and resulting center of mass acceleration. Different imposed delays (0 ms to 350 ms) were entered into the simulation. Peaks in the cross-correlations emerged at the imposed delay.

---

post-learning delay trials for both the trained and untrained conditions using angular velocity variance, percent time within the limits, and ankle torque SD metrics. Comparisons were made between training groups within each delay trial (i.e., AP-delay/ML-fixed; AP-fixed/ML-delay) using two-tailed paired *t*-tests (Bonferroni corrected).

We initially hypothesized that as participants learned to control their balance with the imposed delays, they would internally predict the delay. Consequently, we expected participants would learn to adjust the temporal relationship between their motor commands and resulting whole-body acceleration in order to maintain a near zero delay between these variables. To explore this possibility, we first used a Linear Quadratic Regulator (LQR)

to control standing balance in the presence of a predictor that is able to account for the effects of the delay (see Fig. 9a). We modeled body dynamics as an inverted pendulum (m = 63.11 kg, I = 62.91 kg·m²) with the passive stiffness set as 70% of the load stiffness[88] while the damping ratio was set at 5.73 Nm·s/rad[14]. To replicate the physiological control of standing balance, we added noise on the angular position, angular velocity and torque of the simulation. We modeled angular position and angular velocity noises as pink noise based on the perceptual thresholds for balance perturbations (RMS values of 0.001 rad and 0.003 rad/s for angle and angular velocity noises, respectively)[26,91]. A signal-dependent noise was added to the torque output from the controller (see available code[92]) based on the amplitude of

the exerted torque[93]. We further implemented physiological delays by including a sensory delay ($\Delta t_S = 60$ ms), a motor delay ($\Delta t_M = 60$ ms), and an artificial robotic delay (up to $\Delta t_R = 350$ ms; see below). To solve the control problem, we obtained the optimal solution that minimized the following cost function

$$J = \int_0^\infty [x^T(t)Qx(t) + u^T(t)Ru(t)]dt \qquad (2)$$

Where Q is a positive semi-definite matrix that weighs the states, R is a positive definite matrix that weighs the control efforts (i.e., torque), x is the vector of states of the inverted pendulum (position and velocity), and u is the control effort applied to the pendulum. Using the continuous-time algebraic Riccati equation, we solved the control problem under the cost function J and obtained the gains to control the pendulum given the uncertainties in controlling the system caused by noise and delays. We obtained the gains of the controller by representing imposed and learned delays (see below) as Padé approximations. To test our hypothesis that the timing between motor actions and resulting whole-body acceleration would be adjusted to maintain standing balance with imposed delays up to 350 ms (i.e. replicating experimental conditions), we incorporated a Smith predictor[94] in the control to anticipate the consequences of motor actions in the presence of delays and subsequently regulate the system. Our Smith predictor was comprised of a plant model, a delay model, and a state predictor model. The plant and delay models are exact representations of the plant and all delays. The state predictor model simulated the dynamics of the inverted pendulum (i.e., plant model) and LQR controller using the state after the sensory delay as input, and recursively estimated the torque and states of the system attained forward in time by a period equal to a learned delay or $\Delta t_L$. $\Delta t_L$ represents a period of time equal to the delays that the simulation had to accommodate in order to maintain an upright posture. Under normal circumstances, $\Delta t_L$ is the sum of the motor and sensory delays ($\Delta t_S + \Delta t_M$). When a robotic delay was imposed, $\Delta t_L$ was the sum of all delays ($\Delta t_S + \Delta t_M + \Delta t_R$), whereby the simulation control scheme had fully compensated for the imposed delay. We then compared the predicted state with the set point to provide an error with respect to the desired state forward in time at $t = t^* + \Delta t_L$ (see comparison 1). Note that $t^*$ is defined as the time when the torque enters the plant, resulting in both an angle and angular velocity of the pendulum. We also compared the noisy state with the controller-predicted delayed states in order to estimate the error that was synchronous with the actual feedback at time $t = t^* - \Delta t_S$ (see comparison 2). An additional gain term (k; value < 1, specific for each delay) was placed on this second error to ensure that the set-point error was dominant[95]. These two errors were then added and fed as input to the LQR controller (Fig. 9a), which in turn formulated an action for the present state, optimized for time $t = t^* + \Delta t_M + \Delta t_R$.

We then simulated the inverted pendulum's control of balance using the LQR controller with our Smith predictor ($n = 6$ repetitions) for imposed delays ($\Delta t_R$) ranging from 0 ms to 350 ms (0, 100, 200, 250, 300, 350 ms). These simulations replicated conditions with a natural delay (i.e., $\Delta t_S + \Delta t_M$) and increased up to the delays imposed in the current experiments. The results from the simulations showed that the controller with Smith predictor could maintain the pendulum upright for learned delays up to 470 ms (i.e., $\Delta t_R = 350$ ms; see Fig. 9b insets). We then examined the temporal relationship between the torque applied to the pendulum and whole-body acceleration by estimating the cross-correlation between these two signals. We expected that the cross-correlation would reveal an internal prediction of the delays, such that the peak correlation would occur at time zero once the model had accommodated the delay. Instead, the cross-correlation analyses revealed that the peak positive correlation shifted from $2 \pm 0$ ms for the 0 ms delay, to $-346 \pm 4$ ms for the 350 ms delay (see Fig. 9b). Hence, contrary to our hypothesis, the controller with a predictor did not adjust the timing of the motor commands with respect to whole-body acceleration in order to balance the inverted pendulum. This was because the imposed robotic delay ensured that the whole-body acceleration was always behind in time relative to the generated torque (see Fig. 9a). The

experimental data yielded similar observations and are not presented in the manuscript because the simulation results demonstrate that relative shifts in torque-acceleration timing are not a main feature of delay prediction in the controller.

As an alternative approach to assessing changes in balance control behavior due to delayed training, we analyzed the spectral properties of participant-generated ankle torque and CoM angular velocity. Here, we expected that as participants trained with the imposed delay, they would reduce the power in their ankle torques and angular velocity. Furthermore, this analysis allowed us to assess whether the frequency content of ankle torque and sway velocity changed throughout training. To explore these questions, we estimated the auto spectra of ankle torque and sway velocity signals from the training trials. For each participant, we extracted data in non-overlapping 5 s windows (providing a frequency resolution of 0.2 Hz) when participants remained within the simulated balance limits. Data extracted were limited to multiples of 5 s, such that if there was a 11-second segment of continuous balance, only the first two 5 s windows (i.e., first 10 s) were extracted. We defined the first 10 extracted segments of 5-second windows (i.e., 50 s of data) as the start of training, and the last 10 extracted segments as the end of training. For each participant, we then calculated the average auto spectra across these segments in the frequency domain (see Figs. 2d, 3d). To assess any changes in the frequency distribution of the signals, we also normalized the auto spectra by their sum over a frequency band of 0.2–5 Hz (see Figs. 2d, 3d insets). Comparison between the start and end of training was made by plotting the group averages together with bootstrapped 95% confidence intervals. The 95% confidence intervals for the start and end of training were estimated separately by resampling the data, drawing 12 random subject responses with replication from the sample 10,000 times. Finally, to assess whether training with delays resulted in any aftereffects in normal balance, we then examined the frequency content of ankle torque and sway velocity in the pre- and post-training baseline trials (i.e., AP-baseline/ML-fixed, AP-fixed/ML-baseline, and AP-baseline/ML-baseline). Non-normalized autospectra of ankle torques and sway velocity in these trials were estimated with the same approach as above using both repetitions of the baseline trials (i.e., 60 seconds of data). Similar to the training trials, group means and bootstrapped 95% confidence intervals of pre- and post-learning baseline trials were plotted together to identify any changes in the frequency distribution.

**Experiment 2**. For Experiment 2, we used pairwise comparisons (two-tailed paired $t$-tests) to determine whether biomechanical interactions influenced whole-body standing behavior between the AP and ML directions. First, we tested whether imposing a delay in one direction of standing balance control would influence whole-body motion in the orthogonal axis. To test if an ML-delay increased the variability of AP standing, we compared AP angular velocity variance between the AP-baseline/ML-baseline vs AP-baseline/ML-delay conditions. Conversely, to test whether an AP delay increased the variability of ML standing, we compared ML angular velocity variance between the AP-baseline/ML-baseline vs AP-delay/ML-baseline conditions. We further performed additional pairwise comparisons between different conditions to establish whether these biomechanical interactions influenced whole-body standing behavior. To assess whether angular behavior in baseline conditions differed between trials with motion restrained to one direction or free in both directions, we compared AP angular velocity variance in the AP-baseline/ML-fixed vs AP-baseline/ML-baseline conditions and ML angular velocity variance in the AP-fixed/ML-baseline vs AP-baseline/ML-baseline conditions. Additionally, we examined whether angular behavior in the imposed delay direction differed when standing freely in both directions or with motion restrained to a single direction. This was tested by comparing AP angular velocity variance in the AP-delay/ML-fixed vs AP-delay/ML-baseline conditions and ML angular velocity variance in the AP-fixed/ML-delay vs AP-baseline/ML-delay conditions. Finally, we examined whether the destabilizing effects of a delay differed when imposing delays simultaneously in both directions of standing.

**Article**

This was tested by comparing AP angular velocity variance from AP-delay/ML-baseline vs AP-delay/ML-delay conditions and ML angular velocity variance from AP-baseline/ML-delay vs AP-delay/ML-delay conditions. All comparisons were performed using paired t-tests and the p-value threshold was corrected using a Bonferonni correction for multiple comparisons.

**Experiment 3**. For Experiment 3, all analyses were run on features of whole-body balance movement (angular velocity variance and the percentage of time within the limits) to ensure that comparisons were equivalent across both hand and leg balance conditions. To quantify the learning rate during hand control training trials on a participant-by-participant basis, we fitted first-order exponential functions to the velocity variance and the percentage of time within the limits data obtained over the 18 min of training. We expected that training with the imposed AP-delay during hand control would reduce angular velocity variance and increase the percent time within the balance limits. We used pairwise comparisons with one-tailed paired t-tests to compare balance behavior in the pre- vs post-learning hand control trials. To test our hypothesis that balance improvements gained from hand AP training would transfer to leg AP and leg ML control, we used one-tailed t-tests to compare balance behavior in pre- vs. post-learning for the leg AP-delay and leg ML-delay conditions.

### Reporting summary
Further information on research design is available in the Nature Portfolio Reporting Summary linked to this article.

### Data availability
The source Matlab data that support the findings of this study and generate the main figure results are available in "Data and code for "Learning to stand with sensorimotor delays generalizes across directions and from hand to leg effectors"" https://doi.org/10.34894/AT8YSZ, DataverseNL[92]. Datasets generated and analyzed during the study are available from the corresponding author upon reasonable request.

### Code availability
The source Matlab code used to generate the main results are available in "Data and code for "Learning to stand with sensorimotor delays generalizes across directions and from hand to leg effectors"" https://doi.org/10.34894/AT8YSZ, DataverseNL[92].

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

## Acknowledgements

This study was funded through the Dutch Research Council (NWO) NWO Talent Programme Vidi awarded to Patrick A. Forbes (VI.Vidi.203.066). Brandon G Rasman received funding from the Dutch Research Council (NWO) through the Research Talent Program (406.18.511). Jean-Sébastien Blouin received funding through the Natural Sciences and Engineering Research Council of Canada (RGPIN-2020-05438).

## Author contributions

Brandon G Rasman, Conceptualization, Methodology, Software, Formal analysis, Investigation, Visualization, Project administration, Writing – original draft, Writing – review and editing; Jean-Sébastien Blouin, Conceptualization, Methodology, Visualization, Funding acquisition, Writing – review and editing; Amin M. Nasrabadi, Formal analysis, Writing – review and editing; Remco van Woerkom, Investigation, Methodology, Writing – review and editing; Maarten A Frens, Conceptualization, Visualization, Writing – review and editing; Patrick A Forbes, Conceptualization, Methodology, Resources, Software, Formal analysis, Supervision, Funding acquisition, Investigation, Visualization, Project administration, Writing – review and editing.

## Competing interests

The authors declare no competing interests.
