## [Peer Review File · Communications Biology]

Reviewers' comments:

Reviewer #1 (Remarks to the Author):

This is a well written paper with nice figures reporting a very thorough study. It questions the ability of humans to transfer the learned ability to balance upright with imposed delays across orthogonal space and biomechanically independent muscle effectors engaged in balance control. The answer is that it is indeed the case, the transfer can take place. Although it is very interesting, this conclusion was not as unexpected as stated by the authors, and maybe related to the very specific context of the protocol used to test the transfer.

Also, the authors state in the introduction that the observed ability to generalize sensorimotor learning to delays remains inconsistent; during upper-limb reaching behaviors, transfer of motor learning with delays would be primarily limited or absent, and rightly emphasize that it is often influenced by the contextual similarities of the tasks. This contextual aspect should be more documented in the introduction by detailing the previous results on that topic, which are instructive. Just to quote one example, the Brenner paper conclusion in 2014 stated that that people readily learn to cope with delayed feedback and that such adaptation transfers to new circumstances within what could be considered to be the same task, but does not transfer to completely different tasks. These authors interpret this to mean that subjects learn to control the item that determines their success within a certain context rather than generally adapting to a changed temporal relationship between their motor commands and the sensory consequences or learning precisely how to move in specific circumstances. Such context- dependent learning could explain why we can so easily learn to deal with various tools, even if they introduce delays. As they rightly point out, defining a "task" within this context is far from straightforward. It is very clear and it looks like the present study is one more confirmation of this pioneering hypothesis.

Indeed, following the logic of the Brenner paper, it would be expected, that the nervous system is capable of generalizing learning to balance with imposed delays across the different movement directions contributing to upright because it occurs in the same context. And not surprisingly it is the case. Indeed, this paper demonstrates that humans can transfer the learned ability to balance upright with imposed delays across orthogonal space and biomechanically independent muscle effectors engaged in balance control. However, this is not the interpretation of the authors and it is not clear why. They state that their results reveal that the human brain can leverage prior experience of balancing with sensorimotor delays to control upright posture in vastly different contexts. Why they define their protocol as exploring vastly different context is not clear. This assumption, as pointed by Brenner study is entirely based on how to define "task" or hear a "context". Looks that taking care of delays occurring in postural control in the same complex robotic balance simulator although in different planes could be equally well be considered as taking place in the same context (same simulator, same task ie postural control). Hence it could be said that the present paper merely confirms the conclusion of the Brenner paper in 2014: the similarities and dissimilarities of the context will be based both on objective differences and the subjective appreciation of the subject. This complex mixture it is going to regulate the transfer of any motor learning and as pointed in the 2014 paper defining a "task" within or a "context" is far from straightforward.

In other word the transfer of motor learning may vary with every task or context tested and according to the subjective opinion of each subject on the similarities or dissimilarities of these tasks and/or context. It will depend of his/her training, his/her anxiety if the task is difficult etc. as well describe in the discussion on the debrief session. This point is only examined in the five last lines of the paper when the authors discuss explicit and implicit learning.

Also, it is assumed that the transfer across the planes reveal a beneficial and basic feature of postural

control which is not totally obvious. The protocol is highly sophisticated and far from being ecological. The end result of the transfer could also be considered as a transfer of noise in motor control when facing a highly destabilizing experience both at the biomechanical and cognitive level.

Detailed remarks

" Failing to accommodate for these different sensorimotor delays in balance control increases the risk of falling." It is reasonable but unproven to my knowledge. This should be presented as an hypothesis or papers should be quoted.

In previous studies, the maximum sensory conduction velocity was about 55 m/sec. The velocity in the slowest components of the sensory potentials averaged 20 m/sec. How the authors can exclude that by modulating the nerve conduction velocities, the delays can be compensated ? Please explain;

Participants could quickly transition from sustained periods of balancing with long imposed delays to baseline standing without imposed delays. Could the authors explain why there no after-effect as it is usually the case after any sensorimotor adaptation. One explanation would be again that the transfer take place at a high cognitive level and that the context was indeed recognized as similar, contrary to what happens to more basic "low level" adaptation ad for instance the VOR adaptation. Please discuss that point.

When participants stood freely in both AP and ML directions with a 200 ms delay imposed in only one direction, standing balance became more variable in both directions. The authors state that it reveals the contribution of biomechanical interactions between standing directions and demonstrates that an imposed delay in one axis destabilizes both directions of motion. The influence of these biomechanical interactions on multidirectional standing balance would suggest that they may be, at least partially, responsible for the transfer of training benefits observed in Experiment 1. I do fully agree with these two conclusions. One could argue also for instance that the postural control in the roll and pitch plane being not entirely independent, any noise in a given plane would be observed also in another plane both at the biomechanical AND neural level. For instance, it may be impossible to introduce a neural delay in the neural network controlling the roll plane without contributing to introduce the same delay in the pitch plane. Also, unspecific influence such as arousal, anxiety, selective attention could be at play. The fact that the destabilizing effects of a delay do not differ when imposing delays in a single direction or simultaneously in both directions could support these hypotheses.

The results of experience 3 demonstrate that learning to balance the whole-body with delays using hand muscles can be generalized to untrained control of balance with delays using leg muscles. It confirms that the brain can generalize learned control policies of balancing with sensorimotor delays across biomechanically independent muscle effectors. Therefore, the authors conclude that the brain generalizes learned control policies of balancing with long sensorimotor delays across movement directions and biomechanically independent motor effectors. On the other hand, it is not clear why their first hypothesis derived from experience 1 is not equally plausible: biomechanical interactions across directions may promote transfer because the nervous system has already adjusted the control of certain muscles contributing to the multidirectional control of standing balance. Why these two explanations would be mutually exclusive and not context and person dependent during motor learning?

Reviewer #2 (Remarks to the Author):

This manuscript follows up on a previous study from the same team (<https://doi.org/10.7554/eLife.65085>) where they used a robotic platform to simulate delays between motor actions - in this case, balancing with the ankles - and the resulting sensory feedback - the actual angle of the standing position. While this initial study provided some evidence that human participants could learn to balance with significant delays, the present work explores how such learning could generalise across controlled dimensions and effectors.

In a series of three experiments, they here show that (i) learning to balance in a single dimension (eg antero-posterior) with delays generalises in the orthogonal plane (eg. medio-lateral), but (ii) delays in one dimension affect control in both planes, suggesting high mechanical interconnection, still (iii) learning to balance with delays using a hand trigger (ankles being fixed) also generalises to ankle control.

While I do get the general message, I must confess I struggled a bit with this manuscript due to the lack of details. Methods and rationale are a bit superficial at times, scarce in equations, the terminology is not always clear, and figures are in my opinion incomplete. Also, while my expertise includes motor control (optimal theory), I am not from an engineering background and I found it difficult in particular to follow what is exactly the control policy of the platform, which is a shame if this piece is intended for a more general public.

My main concern however is the general framing of the paper: although adding delays is indeed the experimental manipulation used to disturb balance, it seems that participants do not really learn to integrate those delays in their motor policies but rather use a satisficing strategy (reducing the motor gain), likely coming from a high level, cognitive process (fast, explicit, no after effects, etc.). Such coping strategies are usually trivially generalizable as they relate more to cognition than motor systems. I do appreciate that the authors are very careful about this point in the introduction and also have some elements of reflection in their discussion, but still, I feel that by not properly defining the type of learning occurring before asking about its generalizability, they are putting the cart before the horses and are somehow misleading the reader. Similarly, I wonder if the effect reported is really specific to delays: wouldn't another type of perturbation altering the motor output (eg. adding random noise to the torque) yield similar results, ie some reduction of the motor gain, as seen usually when learning to control new mechanics? Not being so familiar with the literature about balance control, and while I see the potential of this study, I still feel I lack some better context to really appreciate its real implication.

Specific points:

- I cannot but wonder about the reverse question concerning the reported learning behaviour: can people learn to control in the presence of delays in one direction without affecting their policy in the other direction? For example, if participants were exposed to an AP-delay/ML-baseline session like in experiment 2 but for a longer duration allowing for learning. Critically, this could reveal if learning can reduce movement variance in the delayed dimension without affecting the gain and frequency of the intermittent control in the orthogonal dimension, arguing in favour of a genuine motor learning process. If not, that is if the participant cannot overcome the biomechanical interactions, then the learning process is more likely operating at a higher level and the "generalization" of the motor behaviour is, in fact, a misnomer (one does not generalise if elements are not dissociable in the first place).

- I cannot wrap my mind around the time shifting used to analyse experiment 1. In the legend, the authors say that the shift allows to align the torque with the expected load-stiffness relationship, suggesting that they display the torque happening 350ms *after* the measured angle, which kind of makes sense. However, in the main text, they say the shift aligns the torque to the "resulting" body sway (which I understand here as the angular position), suggesting that they display the torque happening *before* the measured angle. Which one is it?

On the same topic, I do not follow the rationale for expecting the 0-shifted relationship to better predict the observed post-learning behaviour. Could the author please clarify how they derived this

prediction? Motor control is usually formalised as a dynamic interplay between prediction, perception, and action, so I wouldn't expect an optimal motor strategy integrating delays to be simply shifted in time. The authors should better describe how they formed this assumption: which simulation results or assumptions were used, etc.

- The equation of the inverted pendulum states that the torque to apply at the ankle to bring the body back to the vertical is a function of both the angular position (to compensate for gravity) and the angular acceleration (to compensate for the inertia of ongoing motion). However, the "reference policy" used to assess learning is based only on the former ("load stiffness", the definition of which should be provided next to the first occurrence in the manuscript). As participants are more unstable (higher accelerations) before learning, the increase in torque variance shown in Fig.3 might in fact be partly expected from the optimal control policy but is presented as an error to be reduced through learning. In other words, plots and sufficient statistics (torque error) seem spuriously inflated by ignoring the acceleration term. If my reasoning is correct, then I would suggest plotting the reference torque as a function of both angle and angular acceleration, and then comparing the actual behaviour to this reference.

- Balancing often takes the form of an intermittent control, as mentioned by the authors. It would be extremely insightful to report how the motor behaviour changes with learning from this perspective: is it the gain or the frequency of the controlling torque which supports adaptation?

- "sway" is defined (at least in my dictionary) as a rhythmical back-and-forth movement. Here, it seems to be used to refer either to the movement (angular velocity) or the angular position, sometimes something in between ("sway position" and "sway velocity"). Therefore, I sometimes found it hard to follow if the authors refer to position, velocity, or something else when using this word.

- It seems that only a subpart of the results are reported. For example, experiment 1 consists of 5 measures which are tested pre/post. However, I see only one measure reported for each group (Figure 2). First, as some conditions serve as controls, they need to be plotted and statistically tested. Otherwise, why were there included in the first place? The problem is particularly annoying in Fig5 (not mentioning the lack of axis legends, reference point in A, the legend on the way...) where not even the comparisons reported in the main text are shown.

- I.204: "though not to the same extent as in the trained direction". This statement needs to be supported by a proper statistical analysis (interaction term of the ANOVA).

- Having a visual representation of the timeline and design of each experiment (What are the pre and post conditions? What is trained and for how long?, etc) would be extremely helpful.

- Please make sure that ALL axes are named (eg. figure 2A), that experience and conditions are explicitly written, and that a proper legend is included in all figures (eg what does the yellow mean in Fig1E?).

Response to Reviewers

Below we list the Reviewers' comments and are our responses. The responses are written in blue text.

Reviewers' comments:

Reviewer #1 (Remarks to the Author):

This is a well written paper with nice figures reporting a very thorough study. It questions the ability of humans to transfer the learned ability to balance upright with imposed delays across orthogonal space and biomechanically independent muscle effectors engaged in balance control. The answer is that it is indeed the case, the transfer can take place. Although it is very interesting, this conclusion was not as unexpected as stated by the authors, and maybe related to the very specific context of the protocol used to test the transfer.

We thank the reviewer for their positive comments about our paper and appreciate the literature highlighted by the reviewer on these topics. We believe there are valid reasons to hypothesize either that (1) learned balance control will not transfer across orthogonal space and biomechanically independent muscle effectors (our null hypothesis presented in the Introduction and Fig. 1), or that (2) learned balance control will transfer across orthogonal space and biomechanically independent effectors (our alternative hypothesis presented in the Introduction and Fig. 1). We have elaborated further on these possibilities below and have amended the manuscript to better highlight these two outcomes. In addition, we have placed further emphasis on our attempt to control for the confound of biomechanical interactions while examining the presence of this potential transfer. Note that we have separated the reviewer's main response into different sections to simplify our responses to specific points.

Also, the authors state in the introduction that the observed ability to generalize sensorimotor learning to delays remains inconsistent; during upper-limb reaching behaviors, transfer of motor learning with delays would be primarily limited or absent, and rightly emphasize that it is often influenced by the contextual similarities of the tasks. This contextual aspect should be more documented in the introduction by detailing the previous results on that topic, which are instructive. Just to quote one example, the Brenner paper conclusion in 2014 stated that that people readily learn to cope with delayed feedback and that such adaptation transfers to new circumstances within what could be considered to be the same task, but does not transfer to completely different tasks. These authors interpret this to mean that subjects learn to control the item that determines their success within a certain context rather than generally adapting to a changed temporal relationship between their motor commands and the sensory consequences or learning precisely how to move in specific circumstances. Such context- dependent learning could explain why

we can so easily learn to deal with various tools, even if they introduce delays. As they rightly point out, defining a “task” within this context is far from straightforward. It is very clear and it looks like the present study is one more confirmation of this pioneering hypothesis.

Indeed, following the logic of the Brenner paper, it would be expected, that the nervous system is capable of generalizing learning to balance with imposed delays across the different movement directions contributing to upright because it occurs in the same context. And not surprisingly it is the case. Indeed, this paper demonstrates that humans can transfer the learned ability to balance upright with imposed delays across orthogonal space and biomechanically independent muscle effectors engaged in balance control. However, this is not the interpretation of the authors and it is not clear why. They state that their results reveal that the human brain can leverage prior experience of balancing with sensorimotor delays to control upright posture in vastly different contexts. Why they define their protocol as exploring vastly different context is not clear. This assumption, as pointed by Brenner study is entirely based on how to define “task” or hear a “context”. Looks that taking care of delays occurring in postural control in the same complex robotic balance simulator although in different planes could be equally well be considered as taking place in the same context (same simulator, same task ie postural control). Hence it could be said that the present paper merely confirms the conclusion of the Brenner paper in 2014: the similarities and dissimilarities of the context will be based both on objective differences and the subjective appreciation of the subject. This complex mixture it is going to regulate the transfer of any motor learning and as pointed in the 2014 paper defining a “task” within or a “context” is far from straightforward.

We thank the reviewer for the positive remarks about our paper and for raising these points. We agree that the ideas and hypotheses proposed by de la Malla et al. ¹ are relevant to our study. However, we also argue that our study is a significant step beyond confirming the findings and/or hypothesis of de la Malla et al. As pointed out in the conclusion of the de la Malla et al. 2014 paper, *“defining a ‘task’ within this context is far from straightforward, so we are aware that a lot remains to be done before we will really understand why transfer occurs in some cases and not in others”*. We agree with this statement. Across the field of motor control, motor learning can be observed to generalize, partially generalize, or show no generalization. There are a variety of studies demonstrating each case, with an incomplete understanding of how and why learning generalizes. This is also the case specific to learning novel sensorimotor delays, as there are reports of no transfer^{1,2} or partial transfer³⁻⁵. Therefore, and in particular for standing balance, there remains uncertainty regarding the processes of learning to control movement with these delays, and the possibility (and mechanisms) of generalization. Here, we explored whether generalization of learning occurs across orthogonal space and biomechanically independent motor effectors involved in upright balance control.

In our study, we present two hypotheses: (1) generalization will not occur across orthogonal space and motor effectors (null hypothesis, Fig. 1) or (2) generalization will occur (alternative hypothesis, Fig. 1). We believe there are valid reasons to support both possibilities, giving us the motivation to test which hypothesis is true.

Supporting the first hypothesis, there are significant contextual differences between our tested balance tasks (i.e., leg balance in AP, leg balance in ML, hand balance in AP), making it plausible to predict that generalization will be minimal or absent. These contextual differences relate to the differing sensorimotor factors (i.e., biomechanics, muscle effectors, activation patterns, sensorimotor delays) across tasks. The most dramatic example of these differences exists between the hand balance condition and leg balancing conditions. Balancing with the hand (or index finger) results in a sensorimotor control loop with starkly differing motor action – sensory feedback coupling than traditional leg standing conditions. Additionally, we sway-referenced the ankle-tilt platform (see main manuscript), which limits lower limb proprioception related to ankle angle, cues that are normally available in traditional standing conditions. Finally, there is also evidence from the balance perturbation literature to suggest that generalization will not occur, because participants that adapt to support surface translations in one direction do not transfer this learning to the orthogonal direction⁶. Supporting the second hypothesis of generalization, there is a consistent feature of the overall task structure across all conditions; i.e., *that the participant is required to balance their upright body against gravity by actively modulating motor actions*. This overarching feature supports the possibility that generalization will occur. Given the uncertainty of these outcomes, our study seems well placed to answer these questions.

In addition, our study is uniquely structured to address the role that biomechanical interactions may have in any observed transfer. As we outlined, a neural generalization of learning could facilitate the transfer of training benefits across orthogonal directions of standing balance. However, biomechanical interactions between directions (e.g., muscles generating balancing motor actions along both AP and ML directions) could also contribute. Importantly, these two factors (neural generalization and biomechanical interactions) are not mutually exclusive (i.e., both could simultaneously contribute to observed transfer of training benefits). Experiment 1 (testing generalization across directions of standing balance) was limited by this fact because we cannot attribute the learning to just one factor. Therefore, we took the step to first characterize some of the effects of biomechanical interactions (Experiment 2), and then control for these factors so as to isolate the neural generalization (Experiment 3). Indeed, the transfer of training benefits observed in Experiment 3 allowed us to conclude that the brain can generalize learned control policies of balancing with novel sensorimotor delays across directions and independent muscle effectors.

Taken together, we believe our study addresses novel and important questions regarding sensorimotor learning and generalization to unexpected sensorimotor delays in balance control. Taking the reviewer's comments into account, we have added text (with additional supporting citations) in our Introduction and Discussion to rationalize more concretely our two-hypothesis model and the importance of controlling for the factor of biomechanical interactions. These additions can be found on lines 54 – 94 and 528 – 656.

In other word the transfer of motor learning may vary with every task or context

tested and according to the subjective opinion of each subject on the similarities or dissimilarities of these tasks and/or context. It will depend of his/her training, his/her anxiety if the task is difficult etc. as well describe in the discussion on the debrief session. This point is only examined in the five last lines of the paper when the authors discuss explicit and implicit learning.

We agree with the reviewer that each participant's subjective opinion/experience regarding the similarities and/or dissimilarities of these tasks may influence the amount of learning transfer. The ability for a participant to learn to balance with an imposed delay and generalize learning may be further influenced by the amount a participant consciously recognizes what control policy they need to implement to balance in this novel condition. As the reviewer mentions, this relates to the relative contribution of explicit (i.e., cognitive) and implicit (i.e., subconscious) learning mechanisms involved when learning to balance with imposed delays. Indeed, we presented the results of the debrief sessions to highlight how participants perceived the delayed balancing condition and whether they consciously recognized or employed different control policies to improve their balance performance. In addition, extended frequency analysis of the torque responses in the post-training baseline conditions (see response to reviewer 2) reveals changes in the frequency distribution that are isolated to the direction of the training. While we cannot be certain that participants were or were not consciously aware of the changes in the frequency content of their motor outputs, we suggest that they emerged through implicit processes to counter the effects of the delay. As we note in the revised text, our study was not designed to estimate the separate contributions of explicit and implicit motor learning. Future studies may consider the added benefit of providing participants with explicit knowledge of the delay or instructions for how balance could be best improved.

In our revision, we have expanded our discussion section (lines 591 – 656) on these topics. This section reads:

“One mechanism proposed to stabilize control with long delays is to reduce the magnitude of the motor action needed to respond to a deviation from the desired position (i.e., the controller gain). Indeed, computational models of human standing predict that the balance system decreases feedback control gains (i.e., proportional or derivative gains) to accommodate increased neural delays^{17, 20, 58}. Our results showed partial support for this change in control. First, participants progressively reduced the variability of their ankle-produced torques. Second, in the debrief session, participants often noted that they were able to improve their balance performance by making “calmer” and “smaller” motor actions to control ongoing balance oscillations, changes that should accompany reduced control gains. Adjusting control gains for balance may also facilitate generalization, as it would be suitable for controlling balance with delays in other contexts (e.g., hand vs. leg control). A reduction in sensorimotor gain may further explain why transitioning from delayed-control to baseline balance did not result in obvious destabilizing behavior, because the gain levels acquired for balancing with long delays will be suitable for balancing without delays^{20, 58}. We note, however, that our analysis of torque and whole-body motion are not direct quantifications of feedback gains of the closed-loop

balance controller. Future studies using independent perturbations (mechanical and/or sensory)^{59, 60, 61} may be able to assess the open-loop transfer function between whole-body movement (i.e., sensory feedback) and muscle activity (i.e., motor command) to directly assess changes in the neural controller. Similar approaches may also be used to test alternative hypotheses that the nervous system increases ankle joint stiffness^{58, 62} and/or uses intermittent control policies^{63, 64, 65} to accommodate increased sensorimotor delays in balance control.

Our results also revealed changes in the frequency distribution of torque and whole-body angular velocity during baseline balance after participants had trained with the delay (see Fig. 4, 5 and S3). Specifically, participants generated oscillatory torques at ~1.4 Hz (i.e., an ~700 ms period), ensuring that the torque to whole-body acceleration relationship was inverted relative to normal balance when in the presence of the 350 ms delay (i.e., the half period of 1.4 Hz). Similar shifts in the frequency distribution also occur during arm reaching when adapting to visually induced delays⁵⁶. Notably, this frequency-specific strategy is different from our original expectation (see Materials and methods) that participants would adjust the timing relationship between their motor commands and whole-body motion. Using an LQR model of balance control that predicts the disruptive effects of the delay (see Materials and methods), we showed that the timing between torque and acceleration signals cannot reveal adaptations to the delay because the imposed robotic delay always ensured whole-body motion occurred later than the control torque. Therefore, the production of a 1.4 Hz torque signal may be a strategic adaptation by the nervous system to account for the influence of the delay in a compensatory manner. Despite the frequency changes, its influence on observed balance is relatively minor (i.e., small aftereffect), remaining undetected in the autospectra of the training trials and causing no discernible changes in the overall variability of baseline pre- and post-trials. Exploring the impact of training with different delays on baseline balance may reveal delay-specific changes in control frequencies similar to those observed here.

Aftereffects that emerge after a period of motor learning are often explained to occur through implicit mechanisms of sensorimotor adaptation^{27, 30, 66, 67}, whereby motor output is recalibrated using errors between expected and actual sensory feedback. For example, implicit adaptations are observed in the vestibular control of standing balance, where vestibular-evoked responses are rapidly (dis)engaged and spatially transformed outside of conscious awareness^{21, 22, 24, 68, 69}. Thus, the aforementioned changes in frequency distributions may indicate implicit alterations in the balance controller. On the other hand, the observed aftereffects from learning were small and did not influence overall variability in balance behavior. Therefore, the adaptation to balancing with delays may align better with explicit (i.e., cognitive) learning, where goal attainment (i.e., remaining upright) is dominant and small or absent aftereffects are observed^{30, 70, 71, 72, 73}. The verbal reports from the debrief sessions further imply that participants were aware of changes in their overall whole-body movements and balancing actions, though they rarely perceived them as delayed control. In addition, the observed generalization of learning supports the involvement of cognitive mechanisms because transfer of learning is more pronounced when explicit learning

mechanisms are involved^{30, 71, 74, 75, 76, 77}. Presumably, the net changes in balance control in response to long sensorimotor delays are driven by a combination of implicit (automatic, absent of cognitive processes) and explicit (conscious awareness and cognitive strategies) mechanisms. Although our study was unable to isolate these contributions, future experiments designed for this purpose could contrast learning and generalization results in participants who train with explicit instructions to those that train without.” (lines 591 – 656)

Also, it is assumed that the transfer across the planes reveal a beneficial and basic feature of postural control which is not totally obvious. The protocol is highly sophisticated and far from being ecological. The end result of the transfer could also be considered as a transfer of noise in motor control when facing a highly destabilizing experience both at the biomechanical and cognitive level.

We agree with the reviewer that this protocol is highly sophisticated and challenging to implement in a real-world environment. In particular, future testing is needed to determine whether training on the robotic balance simulator transfers to real-world balance activities. We have added additional text in the discussion addressing this limitation:

“We note, however, that our study is limited as it only assessed standing balance within our robotic simulator. Future studies are needed to determine whether robot-assisted sensorimotor manipulations of the balance control loop, such as those implemented here, translate to improved balance control in everyday activities” (lines 673 – 676).

We further highlight throughout the manuscript (e.g. lines 54 – 57; 120-122; 521 – 526; 678 – 685) that the generalization of learned balance control with long sensorimotor delays across directions of standing balance is beneficial because it reveals that the nervous system can apply learned principles of delayed balance to new conditions. This generalization occurs despite the aforementioned differences in AP vs ML whole-body mechanics, sensory feedback, delays and muscles actuating control. Accommodating for different delays is particularly important because failing to do so increases the risk of falling¹⁸⁻²¹. Our findings are also very different from perturbation-based learning paradigms, where participants adapt their balance control to a support surface perturbation in one direction, but are unable to transfer this learning to the orthogonal direction⁶. Notably, the transfer from hand-control to leg-control may be even more promising, as it could lead to future training protocols where humans can learn new balance tasks using a hand-held controller and transfer this learning to bipedal balance control.

Finally, the reviewer’s comment about a transfer of noise in motor control being observed across our conditions is interesting. Indeed, this may partly explain the cross-over effects we observed in Experiment 2 (no learning experiment), where an imposed delay in balance direction also results in increased angular velocity variability in the orthogonal direction. While we agree that the role of noise in balance motor learning and generalization are worth investigating, it is a topic that is beyond the scope of our current study.

Detailed remarks

“ Failing to accommodate for these different sensorimotor delays in balance control increases the risk of falling.” It is reasonable but unproven to my knowledge. This should be presented as an hypothesis or papers should be quoted.

We have added text to this sentence and provided the relevant references. The new text reads:

“Standing balance represents a motor behavior that would largely benefit from generalizing learned control with delays, given that the human bipedal posture is mechanically unstable^{13, 14, 15, 16} and failing to accommodate for different sensorimotor delays in balance control increases the risk of falling^{17, 18, 19, 20}”(lines 54 – 61)

In previous studies, the maximum sensory conduction velocity was about 55 m/sec. The velocity in the slowest components of sensory potentials averaged 20 m/sec. How the authors can exclude that by modulating the nerve conduction velocities, the delays can be compensated? Please explain.

If we understand the reviewer correctly, the question is whether the balance training in our protocol may have evoked changes in nerve conduction velocities and that such changes could aid in the compensation/adaptation to imposed delays. Indeed, physical training intervention studies have reported increases in nerve conduction velocities and/or shorter neural response latencies following prolonged (often over multiple days or weeks) training²⁶⁻²⁹. Thus, it is plausible to theorize whether the balance training in our study resulted in increased nerve conduction velocities, facilitating a compensation to balance with the imposed delays. However, it is worth noting that observed changes in nerve conduction velocities from physical training are often modest, with response latencies changing on the order of a few milliseconds. Given that our imposed delays in the training experiments were much longer (350 ms for Experiment 1, 300 ms for Experiment 3), we expect any changes in nerve conduction velocities associated with training would only have a modest contribution (~1% of the imposed delay) to accommodating for the imposed delays. Furthermore, in our previous study on humans learning to balance with imposed delays³⁰, we investigated how vestibular contributions to balance are influenced by imposed delays before and after training. We estimated vestibular-evoked muscle responses in the lower leg before, after and ~3 months following training to balance with a 400 ms delay. We observed no changes in the latencies of these vestibular-evoked responses across training periods or imposed delay magnitudes, further suggesting that nerve conduction changes are not a major contributor to the effects observed in the current study.

Participants could quickly transition from sustained periods of balancing with long imposed delays to baseline standing without imposed delays. Could the authors explain why there no after-effect as it is usually the case after any sensorimotor adaptation. One explanation would be again that the transfer take place at a high

cognitive level and that the context was indeed recognized as similar, contrary to what happens to more basic “low level” adaptation and for instance the VOR adaptation. Please discuss that point.

Through additional analyses suggested by reviewer 2 (see below), we are now able to identify changes in baseline control as a result of training with the delay for prolonged periods. Specifically, a 1.4 Hz peak in the control torque emerged that was not observed in pre-training trials (see Fig. 4c and 5c). These changes also only emerged in the direction of training, remaining directionally isolated even when participants were given control in both directions of balance. These changes, however, were relatively small and were not evident in the overall variability of the signals or throughout the training. Because we do not know of any obvious benefits for these changes under baseline conditions, we speculate that this may be a low-level after-effect from motor policy changes to address balancing with the delay. As noted above, to what degree the learning is due to implicit or explicit processes remains outside the scope of this study. Another possibility for the limited impact of the changes is that motor behavior adapted to balance with long delays may occur through gain changes which are not necessarily destabilizing to normal balance conditions. Notably, if one strategy used by participants is to reduce their controller gains (as we suggest in our paper), there is no reason to predict that the lower gain would be destabilizing under normal standing conditions. Indeed, when increasing delays in computational models of human balance, the estimated feedback control gains needed to maintain upright stance remain within the suitable range of gains for normal balance without added delay^{21,31}. We added a sentence in the discussion addressing this point:

The relevant updated section in the discussion now reads:

“One mechanism proposed to stabilize control with long delays is to reduce the magnitude of the motor action needed to respond to a deviation from the desired position (i.e., the controller gain). Indeed, computational models of human standing predict that the balance system decreases feedback control gains (i.e., proportional or derivative gains) to accommodate increased neural delays^{17, 20, 58}. Our results showed partial support for this change in control. First, participants progressively reduced the variability of their ankle-produced torques. Second, in the debrief session, participants often noted that they were able to improve their balance performance by making “calmer” and “smaller” motor actions to control ongoing balance oscillations, changes that should accompany reduced control gains. Adjusting control gains for balance may also facilitate generalization, as it would be suitable for controlling balance with delays in other contexts (e.g., hand vs. leg control). A reduction in sensorimotor gain may further explain why transitioning from delayed-control to baseline balance did not result in obvious destabilizing behavior, because the gain levels acquired for balancing with long delays will be suitable for balancing without delays^{20, 58}. We note, however, that our analysis of torque and whole-body motion are not direct quantifications of feedback gains of the closed-loop balance controller. Future studies using independent perturbations (mechanical

and/or sensory)^{59, 60, 61} may be able to assess the open-loop transfer function between whole-body movement (i.e., sensory feedback) and muscle activity (i.e., motor command) to directly assess changes in the neural controller. Similar approaches may also be used to test alternative hypotheses that the nervous system increases ankle joint stiffness^{58, 62} and/or uses intermittent control policies^{63, 64, 65} to accommodate increased sensorimotor delays in balance control.

Our results also revealed changes in the frequency distribution of torque and whole-body angular velocity during baseline balance after participants had trained with the delay (see Fig. 4, 5 and S3). Specifically, participants generated oscillatory torques at ~1.4 Hz (i.e., an ~700 ms period), ensuring that the torque to whole-body acceleration relationship was inverted relative to normal balance when in the presence of the 350 ms delay (i.e., the half period of 1.4 Hz). Similar shifts in the frequency distribution also occur during arm reaching when adapting to visually induced delays⁵⁶. Notably, this frequency-specific strategy is different from our original expectation (see Materials and methods) that participants would adjust the timing relationship between their motor commands and whole-body motion. Using an LQR model of balance control that predicts the disruptive effects of the delay (see Materials and methods), we showed that the timing between torque and acceleration signals cannot reveal adaptations to the delay because the imposed robotic delay always ensured whole-body motion occurred later than the control torque. Therefore, the production of a 1.4 Hz torque signal may be a strategic adaptation by the nervous system to account for the influence of the delay in a compensatory manner. Despite the frequency changes, its influence on observed balance is relatively minor (i.e., small aftereffect), remaining undetected in the autospectra of the training trials and causing no discernible changes in the overall variability of baseline pre- and post-trials. Exploring the impact of training with different delays on baseline balance may reveal delay-specific changes in control frequencies similar to those observed here.

Aftereffects that emerge after a period of motor learning are often explained to occur through implicit mechanisms of sensorimotor adaptation^{27, 30, 66, 67}, whereby motor output is recalibrated using errors between expected and actual sensory feedback. For example, implicit adaptations are observed in the vestibular control of standing balance, where vestibular-evoked responses are rapidly (dis)engaged and spatially transformed outside of conscious awareness^{21, 22, 24, 68, 69}. Thus, the aforementioned changes in frequency distributions may indicate implicit alterations in the balance controller. On the other hand, the observed aftereffects from learning were small and did not influence overall variability in balance behavior. Therefore, the adaptation to balancing with delays may align better with explicit (i.e., cognitive) learning, where goal attainment (i.e., remaining upright) is dominant and small or absent aftereffects are observed^{30, 70, 71, 72, 73}. The verbal reports from the debrief sessions further imply that participants were aware of changes in their overall whole-body movements and balancing actions, though they rarely perceived them as delayed control. In addition, the observed generalization of learning supports the involvement of cognitive mechanisms because transfer of learning is more pronounced when explicit learning mechanisms are involved^{30, 71, 74, 75, 76, 77}. Presumably, the net changes in balance

control in response to long sensorimotor delays are driven by a combination of implicit (automatic, absent of cognitive processes) and explicit (conscious awareness and cognitive strategies) mechanisms. Although our study was unable to isolate these contributions, future experiments designed for this purpose could contrast learning and generalization results in participants who train with explicit instructions to those that train without.” (lines 591 – 656)

When participants stood freely in both AP and ML directions with a 200 ms delay imposed in only one direction, standing balance became more variable in both directions. The authors state that it reveals the contribution of biomechanical interactions between standing directions and demonstrates that an imposed delay in one axis destabilizes both directions of motion. The influence of these biomechanical interactions on multidirectional standing balance would suggest that they may be, at least partially, responsible for the transfer of training benefits observed in Experiment 1. I do fully agree with these two conclusions. One could argue also for instance that the postural control in the roll and pitch plane being not entirely independent, any noise in a given plane would be observed also in another plane both at the biomechanical AND neural level. For instance, it may be impossible to introduce a neural delay in the neural network controlling the roll plane without contributing to introduce the same delay in the pitch plane. Also, unspecific influence such as arousal, anxiety, selective attention could be at play. The fact that the destabilizing effects of a delay do not differ when imposing delays in a single direction or simultaneously in both directions could support these hypotheses.

We agree with the reviewer that our results from Experiment 2 support the possibility that balance control in the anteroposterior (i.e., pitch) and mediolateral (i.e., roll) directions are not biomechanically independent. We also agree that control over these two directions may also interact at a neural level, as has been suggested by others^{25,32,33}. Interestingly, we found directionally isolated effects of changing torque frequencies in baseline conditions following training, implying that these small control changes can remain isolated to each direction. We emphasize these points regarding interactions between both directions of balance throughout the revised manuscript (see changes in Introduction, Results, Discussion and Methods sections). We also appreciate the reviewer’s point that factors such as arousal, anxiety and selective attention could also play a role, but we refrain from speculating on these factors since we did not design the experiments to measure these factors.

The results of experience 3 demonstrate that learning to balance the whole-body with delays using hand muscles can be generalized to untrained control of balance with delays using leg muscles. It confirms that the brain can generalize learned control policies of balancing with sensorimotor delays across biomechanically independent muscle effectors. Therefore, the authors conclude that the brain generalizes learned control policies of balancing with long sensorimotor delays across movement directions and biomechanically independent motor effectors. On the other hand, it is not clear why their first hypothesis derived from experience 1 is not equally plausible: biomechanical interactions across directions may promote

transfer because the nervous system has already adjusted the control of certain muscles contributing to the multidirectional control of standing balance. Why these two explanations would be mutually exclusive and not context and person dependent during motor learning?

The reviewer is correct that for the transfer observed in Experiment 1 (transfer of learning across standing balance directions), we are unable to isolate the influence of generalizing learned control policies *and* biomechanical interactions across direction to promote transfer. In presenting these two explanations, we did not intend to suggest that they are mutually exclusive as both can be simultaneously true. We have modified text in our Introduction and Discussion to clarify this point. In Experiment 3, our aim was to control for the factor of biomechanical interactions in order to determine whether biomechanical interactions are required for the transfer of training benefits. Without this control, we would have to make a large assumption to attribute any of the learning to neural generalization.

This logic is presented in our introduction (lines 82 – 94) and our discussion (lines 528 – 555). Notably, the relevant section in the introduction now reads:

“An important consideration for any potential generalization is that lower limb muscles generate joint forces and torques contributing to both balance directions³⁹. As a result, the question arises of whether any transfer of balance improvements is due to a neural generalization of learning or a byproduct of biomechanical interactions (i.e., adapted motor commands of muscles contributing to multidirectional balance control). Importantly, the possibility of neural generalization and biomechanical interactions are not mutually exclusive – i.e., both could contribute to transfer of training benefits. Therefore, to determine whether biomechanical interactions are required for the transfer of training benefits, we assessed whether transfer occurred across biomechanically independent muscles. If the brain can broadly update its control to accommodate for imposed delays, we hypothesized that even when participants trained to balance the whole-body with their hand muscles, balance performance would improve when controlling upright body with postural lower limb muscles.” (lines 82 – 94)

Reviewer #2 (Remarks to the Author):

This manuscript follows up on a previous study from the same team (<https://doi.org/10.7554/eLife.65085>) where they used a robotic platform to simulate delays between motor actions - in this case, balancing with the ankles - and the resulting sensory feedback - the actual angle of the standing position. While this initial study provided some evidence that human participants could learn to balance with significant delays, the present work explores how such learning could generalise across controlled dimensions and effectors.

In a series of three experiments, they here show that (i) learning to balance in a single dimension (eg antero-posterior) with delays generalises in the orthogonal plane (eg. medio-lateral), but (ii) delays in one dimension affect control in both planes, suggesting high mechanical interconnection, still (iii) learning to balance with

delays using a hand trigger (ankles being fixed) also generalises to ankle control.

While I do get the general message, I must confess I struggled a bit with this manuscript due to the lack of details. Methods and rationale are a bit superficial at times, scarce in equations, the terminology is not always clear, and figures are in my opinion incomplete. Also, while my expertise includes motor control (optimal theory), I am not from an engineering background and I found it difficult in particular to follow what is exactly the control policy of the platform, which is a shame if this piece is intended for a more general public.

We thank the reviewer for identifying areas of the manuscript that require improvements in clarity. We have addressed these items in the manuscript and provide point-by-point details here about how we addressed these issues. Specific to one point mentioned above, we have also added an additional control diagram to the supplementary material (Fig. S1) and refer in more detail to previous studies, both of which provide more information regarding the control policy of the platform.

My main concern however is the general framing of the paper: although adding delays is indeed the experimental manipulation used to disturb balance, it seems that participants do not really learn to integrate those delays in their motor policies but rather use a satisficing strategy (reducing the motor gain), likely coming from a high level, cognitive process (fast, explicit, no after effects, etc.). Such coping strategies are usually trivially generalizable as they relate more to cognition than motor systems. I do appreciate that the authors are very careful about this point in the introduction and also have some elements of reflection in their discussion, but still, I feel that by not properly defining the type of learning occurring before asking about its generalizability, they are putting the cart before the horses and are somehow misleading the reader. Similarly, I wonder if the effect reported is really specific to delays: wouldn't another type of perturbation altering the motor output (eg. adding random noise to the torque) yield similar results, ie some reduction of the motor gain, as seen usually when learning to control new mechanics? Not being so familiar with the literature about balance control, and while I see the potential of this study, I still feel I lack some better context to really appreciate its real implication.

We appreciate that the reviewer sees the potential of our study. As pointed out above (and as we suggest in our paper), one possible mechanism for the nervous system to accommodate for long imposed delays is to reduce its sensorimotor control gains in balance control. The reviewer suggests that this likely comes from a high level, cognitive process and is therefore trivially generalizable. While we agree that cognitive mechanisms of learning may be at play in our observed learning and generalization, we cannot rule out the contributions of automatic mechanisms. For instance, during standing balance there is evidence to suggest that automatic motor control mechanisms and cognitive perception of upright stance are processed separately, and can arrive at similar or different internal representations of postural state³⁹⁻⁴³. We also point to our previous findings on learning to balance with unexpected sensorimotor delays that indicate that both automatic/subcortical

mechanisms and cognitive mechanisms are involved in the learning process³⁰. Specifically, vestibular contributions to standing balance rapidly attenuate at first exposure to novel delays but recover following prolonged (> 30 min) balance training with the delay³⁰. Given that vestibular contributions to balance are modulated through the unconscious integration of sensory and motor signals of balance (ref⁴¹), it is likely that the changes in our previous study occurred through subcortical mechanisms. Similarly, from an explicitly cognitive perspective, participants who have trained with the delay perceive its effects on balance at a threshold that is ~45 ms longer than before training³⁰. We speculate that the ability to generalize the imposed delays may be affected by both subconscious and conscious processes. Notably, our additional frequency analysis (see below) supports such a possibility.

We also highlight that certain classical motor learning behaviors observed in other motor systems often differ when compared to standing balance. For example, when humans (and animals) are exposed to reversals of vision using head-mounted prisms, vestibulo-ocular reflex responses can be reversed. This adaptation occurs over a period of 1-2 weeks^{44,45}. In contrast, when a similar reversal is imposed on the vestibular control of standing balance⁴⁶ participants are capable of learning to balance under these novel conditions within ~90 s. Importantly, accompanying the rapid learning of this new control state, the vestibular-evoked motor responses become inverted. Thus, unlike traditional expectations of vestibular reflexes, those contributing to the balance system can undergo rapid reassociations of sensory and motor signals that facilitate highly flexible and organized postural responses.

In our revised manuscript, we more clearly discuss the points mentioned above to provide the necessary information to readers less familiar with the balance control field (see Introduction and Discussion). Specific to the reviewer's request for better context regarding the implications of our study, we have revised the Introduction and Discussion to (1) address the contextual differences and similarities across our postural tasks that may inhibit or facilitate generalization, respectively (see major comment from Reviewer 1), (2) provide more details on the mechanisms (both automatic and cognitive) contributing to balance control, which may influence the possibility for generalization, and (3) better emphasize the importance of considering and controlling for biomechanical interactions when interpreting multidirectional postural control and balance learning. We believe that these changes better highlight the broad interest and implications of this study. The relevant changes in the Introduction are as follows:

“Given the likelihood that sensorimotor delays can change^{3, 7, 10, 11, 12}, it would be advantageous if the brain could generalize learned policies for controlling self-motion with delays and transfer them to different contexts. Standing balance represents a motor behavior that would largely benefit from generalizing learned control with delays, given that the human bipedal posture is mechanically unstable^{13, 14, 15, 16} and failing to accommodate for different sensorimotor delays in balance control increases the risk of falling^{17, 18, 19, 20}. However, the generation of multidirectional balance-correcting responses relies on a diversity of sensory signals (i.e., visual, vestibular, somatosensory, auditory) and anatomically distinct muscles (i.e., ankle and hip), which may challenge and limit generalization across directions and muscle effectors.

Imposing long delays into anteroposterior control of human standing destabilizes balance, but through training, participants can learn to regain their upright balance control and retain this ability after three months²¹. This learning is accompanied by modulations in the vestibular control of balance and cognitive perception of self-motion, supporting the view that adaptations to ongoing balance control are governed through both automatic and cognitive processes^{22, 23, 24, 25, 26}. However, whether the nervous system can generalize the learning to different task contexts remains unknown. Typically, the more contextual factors that overlap between tasks (i.e., goal, movement mechanics, sensory cues and motor effectors), the more likely learned control policies will generalize^{27, 28, 29, 30}. Standing balance involves controlling whole-body motion in both anteroposterior and mediolateral space, with each direction of balance possessing distinct biomechanics^{16, 17, 31, 32}, muscle effectors, activation patterns^{33, 34, 35} and sensorimotor delays^{5, 17, 20, 36, 37}. Therefore, these differing sensorimotor factors may limit the ability to generalize learning across orthogonal space (H_0 , see Fig. 1), as observed when standing participants adapt their balance to externally imposed perturbations in different directions³⁸. On the other hand, the common task goal of balancing the upright body against gravity may help facilitate transfer across the different directions of balance (H_1). Here, we explored these two hypotheses to determine whether the learned control policies of balancing with sensorimotor delays generalize across different contexts.

An important consideration for any potential generalization is that lower limb muscles generate joint forces and torques contributing to both balance directions³⁹. As a result, the question arises of whether any transfer of balance improvements is due to a neural generalization of learning or a byproduct of biomechanical interactions (i.e., adapted motor commands of muscles contributing to multidirectional balance control). Importantly, the possibility of neural generalization and biomechanical interactions are not mutually exclusive – i.e., both could contribute to transfer of training benefits. Therefore, to determine whether biomechanical interactions are required for the transfer of training benefits, we assessed whether transfer occurred across biomechanically independent muscles. If the brain can broadly update its control to accommodate for imposed delays, we hypothesized that even when participants trained to balance the whole-body with their hand muscles, balance performance would improve when controlling upright body with postural lower limb muscles.” (lines 51 – 94)

We also highlight the “Clinical implications” section of our discussion. Notably, our study demonstrates that by using a robotic balance simulator that gives the participant an opportunity to train to balance in challenging conditions (i.e., with large imposed delays), one can evoke robust and generalizable learning. This differs from the limited generalization observed in studies that adapt participants to repeated physical perturbations (Harper et al. 2021). Instead, our approach allows the nervous system to explore safely distinct and challenging environments (i.e., novel sensorimotor relationships) and adjust to the related instabilities. Therefore, we believe our findings will be of interest to the clinical and rehabilitative research fields that are focused on designing training therapies that improve balance and transfer to

different circumstances.

Specific points:

- I cannot but wonder about the reverse question concerning the reported learning behaviour: can people learn to control in the presence of delays in one direction without affecting their policy in the other direction? For example, if participants were exposed to an AP-delay/ML-baseline session like in experiment 2 but for a longer duration allowing for learning. Critically, this could reveal if learning can reduce movement variance in the delayed dimension without affecting the gain and frequency of the intermittent control in the orthogonal dimension, arguing in favour of a genuine motor learning process. If not, that is if the participant cannot overcome the biomechanical interactions, then the learning process is more likely operating at a higher level and the "generalization" of the motor behaviour is, in fact, a misnomer (one does not generalise if elements are not dissociable in the first place).

We thank the reviewer for raising this question. Following the reviewer's suggestion from a comment below (i.e., "gain or frequency of controlling torque"), we performed an additional frequency analysis of our learning data that revealed some of the features (dissociation of elements) that the reviewer alludes to in the above comment. Specifically, we compared the frequency content of balance behaviour (angular velocity variability and ankle torque variability) at the start and end of the training with the delay, as well as in pre- and post-learning in baseline conditions. Details of these methods can be found in our new submission (see lines 1102 – 1128). Comparison of the autospectra (i.e., the frequency content) at the start and end of training revealed a decrease in power across the entire frequency bandwidth for both angular velocity and ankle torque (see Fig. 2d and 3d). These results match the decrease in angular velocity and ankle torque variability described in our original submission. Despite this overall change in power, however, we saw no change in the relative distribution of power across frequencies when we normalized the spectra to their sum over a bandwidth of 0.2-5 Hz (see Fig. 2d and 3d insets). This suggests that while participants reduced their torque output (i.e., a proximal estimate of gain) as they progressed through learning, they maintained a consistent control policy throughout.

In contrast, when we compared the baseline balance conditions pre- and post-learning, we saw changes in the frequency distribution of power in both angular velocity and ankle torque signals. Specifically, a peak in autospectral power emerged at ~1.4 Hz and the power at frequencies from 0.4-1.2 Hz decreased. Importantly, these changes in power distribution only occurred in the direction of balance that participants trained with the delay, while power in the orthogonal direction remained relatively unchanged (see Fig. 4c and 5c). This suggests that adaptation of participant motor behavior during balance delay training result in measurable changes in the control of baseline standing when transferring back to normal balance. These changes, however, were restricted to only small redistributions of power across frequencies during baseline standing, and did not emerge during training. We suspect that because the signal power during training is one order of

magnitude larger than baseline conditions (compare Fig. 2d and 3d to Fig. 4c and 5d), it may be difficult to observe these specific changes as participants learn to balance with the delay. Notably, these changes in frequency distribution also failed to influence the overall variance in pre-post comparisons of baseline standing. Directly addressing the reviewer's comment, the observed changes remained isolated to the trained direction even when participants were allowed to balance in both directions during post-training baseline trials (see Fig. S3). Our data imply that the learned control policies needed to account for the delay can be generalized across directions and different motor effectors, but that aftereffects of delay training are small and remain isolated to the trained direction of balance.

- I cannot wrap my mind around the time shifting used to analyse experiment 1. In the legend, the authors say that the shift allows to align the torque with the expected load-stiffness relationship, suggesting that they display the torque happening 350ms *after* the measured angle, which kind of makes sense. However, in the main text, they say the shift aligns the torque to the "resulting" body sway (which I understand here as the angular position), suggesting that they display the torque happening *before* the measured angle. Which one is it?

On the same topic, I do not follow the rationale for expecting the 0-shifted relationship to better predict the observed post-learning behaviour. Could the author please clarify how they derived this prediction? Motor control is usually formalised as a dynamic interplay between prediction, perception, and action, so I wouldn't expect an optimal motor strategy integrating delays to be simply shifted in time. The authors should better describe how they formed this assumption: which simulation results or assumptions were used, etc.

- The equation of the inverted pendulum states that the torque to apply at the ankle to bring the body back to the vertical is a function of both the angular position (to compensate for gravity) and the angular acceleration (to compensate for the inertia of ongoing motion). However, the "reference policy" used to assess learning is based only on the former ("load stiffness", the definition of which should be provided next to the first occurrence in the manuscript). As participants are more unstable (higher accelerations) before learning, the increase in torque variance shown in Fig.3 might in fact be partly expected from the optimal control policy but is presented as an error to be reduced through learning. In other words, plots and sufficient statistics (torque error) seem spuriously inflated by ignoring the acceleration term. If my reasoning is correct, then I would suggest plotting the reference torque as a function of both angle and angular acceleration, and then comparing the actual behaviour to this reference.

We thank the reviewer for raising these three points regarding our timing analysis of ankle-produced torques during delay learning and its relation to both position and acceleration. We have provided a single answer to these three points as they relate to new analysis and simulation work (i.e., first and second point) that has revealed that the timing of the torque-angle and torque-acceleration relationships cannot offer insight into predictive behavior (i.e., the first and third point). In addition, we have

developed a new approach (i.e., the frequency analysis) to improve the analyses of the current data.

We first adopted a more accepted analysis approach to assess the timing of the torque-angle and torque-acceleration relationships using a cross-correlation approach (i.e., `xcorr` in Matlab). This replaced our less conventional method of time-shifting the torque signal and identifying the mean torque deviation (i.e., error) relative to the load-stiffness curve. We found that similar to our original analysis, both metrics showed a shift in time at the start of training that remained unchanged by the end of training (see Fig. R1). In an effort to better understand these responses, we then developed an LQR model that could account for the delay using a Smith predictor. Our aim was to show how a predictive model can facilitate a change in motor control and help explain the responses observed in the human behaviour. Details of the model implementation can be found in the methods section in our new submission (see pages 26-28). Briefly, the Smith predictor simulates the influence of the delay and additional noise on the inverted pendulum in order to allow the controller to compensate for their disruptive effects on balance.

Fig. R1. Cross-correlation estimates between applied ankle torque and center of mass acceleration in pre-training baseline conditions (left) and end of training conditions when balancing with a 350 ms delay (right). Thin lines represent individual participants while the thick lines represent the group averages. In the delay condition, peak correlations emerge at the imposed delay despite the improvement of balance over the 60 minutes of training.

The simulations showed that the controller with Smith predictor could maintain upright balance of the pendulum at imposed delays of 350 ms (see Fig. 8). Using this model, we then examined the timing relationship (i.e., cross-correlation) between torque and acceleration. The cross-correlation analyses revealed that the peak positive correlation shifted from 2 ± 0 ms for the 0 ms delay, to -346 ± 4 ms for the 350ms delay. Similarly, all other shifts in peak correlations matched the imposed delay. Hence, contrary to our expectation, the cross-correlation analysis between torque and acceleration was unable to show that the controller did not adjust the timing of the motor commands with respect to whole-body motion in order to balance the inverted pendulum. This is because the manner in which we imposed the delay (i.e., the torque is first delayed before driving pendulum motion) ensured that the

acceleration was always behind in time (according to the delay) relative to the delivered torque. For demonstration and didactic purposes, we have included the simulation outcomes in the manuscript but have removed the experimental results given that the simulations show that relative shifts in torque-acceleration timing are not a main feature of delay prediction in the controller.

We have also chosen not to use the LQR controller for any further analysis or prediction of the results because we see some discrepancies in the width of the cross-correlation between our simulation and experimental results. Instead, our aim was to simply demonstrate that the cross-correlation analysis (and our time-shifting analysis) cannot be used to reveal changes in timing between motor output and sensory feedback under the current experimental conditions. We are planning future studies to explore how different controllers can account for sensorimotor delays in the control of standing balance.

- Balancing often takes the form of an intermittent control, as mentioned by the authors. It would be extremely insightful to report how the motor behaviour changes with learning from this perspective: is it the gain or the frequency of the controlling torque which supports adaptation?

We thank the reviewer for suggesting the additional frequency analysis of our data. While the details of these analyses are provided in our second main response (see above), we reiterate the main points here. First, we saw a decrease in the magnitude of the frequency content of the control torque throughout training, but these changes did not influence the frequency distribution of power when they were normalized (see Fig. 2d and Fig. 3d). This suggests that participants maintained the dynamic features of control throughout learning and simply decreased their gain to accommodate the delay. Nevertheless, we did see changes in the frequency distribution of control torque when we examined the data in pre- and post-learning baseline conditions. Specifically, a peak in power emerged at a frequency of ~ 1.4 Hz but only in the direction of training (Fig 4c and 5c). This training-direction-specific change in power during baseline post-learning trials was also observed when participants balanced in baseline conditions in both axes (Fig S3). These results indicate that learning to balance with a delay can induce an aftereffect on normal balance once participants transition back to baseline conditions, but that it remains isolated to the direction of the imposed delay and is small relative to the overall balance behaviour. Furthermore, despite these aftereffects only appearing in the trained direction, participants were able to transfer the learned behavior from training in one direction onto the orthogonal untrained direction. In our discussion, we further explored how these changes suggest that implicit (i.e., subconscious) mechanisms, in addition to explicit mechanisms, contribute to learning to balance with imposed delays.

- "sway" is defined (at least in my dictionary) as a rhythmical back-and-forth movement. Here, it seems to be used to refer either to the movement (angular

velocity) or the angular position, sometimes something in between ("sway position" and "sway velocity"). Therefore, I sometimes found it hard to follow if the authors refer to position, velocity, or something else when using this word.

In the field of postural control, "sway" (e.g., whole-body sway, postural sway) is often used to characterize the oscillations of the upright body during standing^{20,24,25,32,40,63-69}. Sway will often refer to the position, velocity and/or acceleration of the body's center of mass, but studies also present sway of different body segments or joint angles^{32,70-74}. Indeed, this can sometimes be confusing and we thank the reviewer for highlighting this.

In our revised submission, we have removed the term "sway" when referring to our data. Instead, we describe angular position, angular velocity and angular acceleration of the whole-body center of mass. We have revised text throughout the document to clarify this distinction (see example lines 161-162).

- It seems that only a subpart of the results are reported. For example, experiment 1 consists of 5 measures which are tested pre/post. However, I see only one measure reported for each group (Figure 2). First, as some conditions serve as controls, they need to be plotted and statistically tested. Otherwise, why were there included in the first place? The problem is particularly annoying in Fig5 (not mentioning the lack of axis legends, reference point in A, the legend on the way...) where not even the comparisons reported in the main text are shown.

We agree with the reviewer that some of the pre/post trials serve as controls and are very important to include in the analysis. Although almost all of these conditions were reported in the original submission, we have revised the manuscript and figures to better highlight these trials. In summary, for Experiment 1, two of the five pre/post conditions (i.e., AP-delay/ML-fixed and AP-fixed/ML-delay) are presented in Fig. 2 and Fig. 3 (see subpanels a, b), and depict the transfer of learning across the different directions of balance. Two other pre conditions (i.e., AP-baseline/ML-fixed and AP-fixed/ML-baseline) are shown in Fig. 2c and 3c (small diamond symbols) for comparison to the training data. These data, together with their equivalent post-learning trials, are shown in Fig. 4 and Fig. 5 in much greater detail. Here we assessed any changes in the overall variability of angular velocity and ankle torque that occurred between pre and post baseline conditions. Regarding the last control condition (i.e., AP-baseline/ML-baseline), the reviewer is correct to point out that this condition was not presented in any figure in the original submission. In the revised submission, we present an identical analysis on both directions of balance in this bi-directional baseline trial. Notably, in addressing this point, we revised the text as follows:

"Similar to the single direction baseline balance trials, comparison of pre vs. post behavior in dual axis control baseline trials (i.e., AP-baseline/ML-baseline) revealed no differences in angular velocity variance and torque SD in both the AP and ML directions (all $p > 0.10$)." (lines 294-297)

In addition, we also presented the frequency analysis of these bi-directional baseline trials in the supplementary information document (Fig. S3). Importantly, the observed

aftereffects in frequency content of angular velocity and ankle torque remained isolated to the trained direction even when participants were allowed to balance in the bi-directional baseline trial.

For Experiment 2, participants performed 8 trials, three of which were baseline without delays while the other five were different combinations of delays and control across the AP and ML axes. In the main body of the manuscript (i.e., Fig 6, previously Fig 5), we compared only two of the delay conditions (AP-delay/ML-baseline and AP-baseline/ML-delay) to a single baseline condition (AP-baseline/ML-baseline) to highlight the primary observation that biomechanical interactions can influence balance when imposing a delay in one axis. Based on the reviewers' comments, we have improved the clarity of this figure by addressing the points raised (i.e., axes legends, reference point in a, legend on the sway).

The results from the remaining conditions are depicted in the supplementary material (see Fig. S5). We chose to separate these data to avoid overloading the reader with the secondary outcomes of these trials. Specifically, these results demonstrate (1) the disruptive effects of the delay in one direction does not depend on the participant being fixed or free in the orthogonal direction, and (2) that imposing delays in both directions of balance evokes destabilizing effects that are comparable to those when imposing a delay in one direction. Despite placing the figure in the supplementary material, we did present the results in the main text to fully characterize the different interaction effects with reference to the supplementary figure (lines 420 – 436).

- I.204: "though not to the same extent as in the trained direction". This statement needs to be supported by a proper statistical analysis (interaction term of the ANOVA).

We thank the reviewer for pointing this out. We have performed additional analyses for Experiment 1 to test for whether relative balance improvement is greater in the trained condition when compared to the untrained conditions. We described these analyses in the methods (see lines 1013 – 1019) and further detail the results in the Supplementary Information document (see Fig. S2).

- Having a visual representation of the timeline and design of each experiment (What are the pre and post conditions? What is trained and for how long?, etc) would be extremely helpful.

We agree with the reviewer and have added a schematic to Fig. 1 that summarizes the balance conditions tested for Experiments 1-3 and the associated timelines for training experiments (Experiments 1 and 3).

- Please make sure that ALL axes are named (eg. figure 2A), that experience and conditions are explicitly written, and that a proper legend is included in all figures (eg what does the yellow mean in Fig1E?).

Thank you. All of the figures, and their captions, have been updated to provide more information. We present legends where necessary and provide further information in figure captions. For instance, in the Fig. 1 caption we describe that the yellow (in the

filled circles) indicates a condition in which delayed balance control is learned through training (i.e., in specific directions or with specific motor effectors).

References

1. de la Malla C, López-Moliner J, Brenner E. Dealing with delays does not transfer across sensorimotor tasks. *J Vis.* 2014;14(12).
2. Rohde M, van Dam LC, Ernst MO. Predictability is necessary for closed-loop visual feedback delay adaptation. *J Vis.* 2014;14(3):4.
3. Avraham G, Leib R, Pressman A, et al. State-Based Delay Representation and Its Transfer from a Game of Pong to Reaching and Tracking. *eNeuro.* 2017;4(6).
4. Cunningham DW, Chatziastros A, von der Heyde M, Bulthoff HH. Driving in the future: temporal visuomotor adaptation and generalization. *J Vis.* 2001;1(2):88-98.
5. Rohde M, Altan G, Ernst MO. When vision lags, motor prediction follows. *bioRxiv.* 2020:2020.2002.2013.937235.
6. Harper SA, Beethe AZ, Dakin CJ, Bolton DAE. Promoting Generalized Learning in Balance Recovery Interventions. *Brain Sci.* 2021;11(3).
7. Mazzoni P, Krakauer JW. An implicit plan overrides an explicit strategy during visuomotor adaptation. *The Journal of neuroscience : the official journal of the Society for Neuroscience.* 2006;26(14):3642-3645.
8. McDougle SD, Bond KM, Taylor JA. Explicit and Implicit Processes Constitute the Fast and Slow Processes of Sensorimotor Learning. *The Journal of neuroscience : the official journal of the Society for Neuroscience.* 2015;35(26):9568-9579.
9. Taylor JA, Krakauer JW, Ivry RB. Explicit and implicit contributions to learning in a sensorimotor adaptation task. *The Journal of neuroscience : the official journal of the Society for Neuroscience.* 2014;34(8):3023-3032.
10. Benson BL, Anguera JA, Seidler RD. A spatial explicit strategy reduces error but interferes with sensorimotor adaptation. *Journal of neurophysiology.* 2011;105(6):2843-2851.
11. Krakauer JW, Hadjiosif AM, Xu J, Wong AL, Haith AM. Motor Learning. *Comprehensive Physiology.* 2019;9(2):613-663.
12. Telgen S, Parvin D, Diedrichsen J. Mirror reversal and visual rotation are learned and consolidated via separate mechanisms: recalibrating or learning de novo? *The Journal of neuroscience : the official journal of the Society for Neuroscience.* 2014;34(41):13768-13779.
13. Yang CS, Cowan NJ, Haith AM. De novo learning versus adaptation of continuous control in a manual tracking task. *eLife.* 2021;10.
14. Mattar AA, Ostry DJ. Generalization of dynamics learning across changes in movement amplitude. *Journal of neurophysiology.* 2010;104(1):426-438.
15. Mauk MD, Buonomano DV. The neural basis of temporal processing. *Annual review of neuroscience.* 2004;27:307-340.
16. McDougle SD, Ivry RB, Taylor JA. Taking Aim at the Cognitive Side of Learning in Sensorimotor Adaptation Tasks. *Trends in cognitive sciences.* 2016;20(7):535-544.
17. Taylor JA, Ivry RB. Flexible cognitive strategies during motor learning. *PLoS computational biology.* 2011;7(3):e1001096.
18. Bingham JT, Choi JT, Ting LH. Stability in a frontal plane model of balance requires coupled changes to postural configuration and neural feedback control. *Journal of neurophysiology.* 2011;106(1):437-448.
19. Kuo AD. An optimal control model for analyzing human postural balance. *IEEE transactions on bio-medical engineering.* 1995;42(1):87-101.
20. van der Kooij H, Jacobs R, Koopman B, Grootenboer H. A multisensory integration model of human stance control. *Biological cybernetics.* 1999;80(5):299-308.
21. van der Kooij H, Peterka RJ. Non-linear stimulus-response behavior of the human stance control system is predicted by optimization of a system with sensory and motor noise. *Journal of computational neuroscience.* 2011;30(3):759-778.

22. Fitzpatrick RC, Taylor JL, McCloskey DI. Ankle stiffness of standing humans in response to imperceptible perturbation: reflex and task-dependent components. *The Journal of physiology*. 1992;454:533-547.
23. Loram ID, Lakie M. Direct measurement of human ankle stiffness during quiet standing: the intrinsic mechanical stiffness is insufficient for stability. *The Journal of physiology*. 2002;545(3):1041-1053.
24. Winter DA, Patla AE, Prince F, Ishac M, Gielo-Periczak K. Stiffness control of balance in quiet standing. *Journal of neurophysiology*. 1998;80(3):1211-1221.
25. Winter DA, Prince F, Frank JS, Powell C, Zabjek KF. Unified theory regarding A/P and M/L balance in quiet stance. *Journal of neurophysiology*. 1996;75(6):2334-2343.
26. Casolo A, Farina D, Falla D, Bazzucchi I, Felici F, Del Vecchio A. Strength Training Increases Conduction Velocity of High-Threshold Motor Units. *Med Sci Sports Exerc*. 2020;52(4):955-967.
27. Del Vecchio A, Negro F, Falla D, Bazzucchi I, Farina D, Felici F. Higher muscle fiber conduction velocity and early rate of torque development in chronically strength-trained individuals. *Journal of applied physiology (Bethesda, Md : 1985)*. 2018;125(4):1218-1226.
28. Gholami F, Khaki R, Mirzaei B, Howatson G. Resistance training improves nerve conduction and arterial stiffness in older adults with diabetic distal symmetrical polyneuropathy: A randomized controlled trial. *Exp Gerontol*. 2021;153:111481.
29. Martinez-Valdes E, Farina D, Negro F, Del Vecchio A, Falla D. Early Motor Unit Conduction Velocity Changes to High-Intensity Interval Training versus Continuous Training. *Med Sci Sports Exerc*. 2018;50(11):2339-2350.
30. Rasman BG, Forbes PA, Peters RM, et al. Learning to stand with unexpected sensorimotor delays. *eLife*. 2021;10.
31. Le Mouel C, Brette R. Anticipatory coadaptation of ankle stiffness and sensorimotor gain for standing balance. *PLoS computational biology*. 2019;15(11):e1007463.
32. Day BL, Steiger MJ, Thompson PD, Marsden CD. Effect of vision and stance width on human body motion when standing: implications for afferent control of lateral sway. *The Journal of physiology*. 1993;469:479-499.
33. Mian OS, Day BL. Violation of the craniocentricity principle for vestibularly evoked balance responses under conditions of anisotropic stability. *The Journal of neuroscience : the official journal of the Society for Neuroscience*. 2014;34(22):7696-7703.
34. Alexander RM. The dimensions of knee and ankle muscles and the forces they exert. *J Hum Move Stud*. 1975;1:115-123.
35. Allen JL, Ting LH. Why Is Neuromechanical Modeling of Balance and Locomotion So Hard? In: Prilutsky BI, Edwards DH, eds. *Neuromechanical Modeling of Posture and Locomotion*. New York, NY: Springer New York; 2016:197-223.
36. Héroux ME, Dakin CJ, Luu BL, Inglis JT, Blouin JS. Absence of lateral gastrocnemius activity and differential motor unit behavior in soleus and medial gastrocnemius during standing balance. *Journal of applied physiology (Bethesda, Md : 1985)*. 2014;116(2):140-148.
37. Ting LH, Chiel HJ, Trumbower RD, et al. Neuromechanical principles underlying movement modularity and their implications for rehabilitation. *Neuron*. 2015;86(1):38-54.
38. Vieira TM, Minetto MA, Hodson-Tole EF, Botter A. How much does the human medial gastrocnemius muscle contribute to ankle torques outside the sagittal plane? *Human movement science*. 2013;32(4):753-767.
39. Dalton BH, Rasman BG, Inglis JT, Blouin JS. The internal representation of head orientation differs for conscious perception and balance control. *The Journal of physiology*. 2017;595(8):2731-2749.
40. Forbes PA, Chen A, Blouin JS. Sensorimotor control of standing balance. *Handbook of clinical neurology*. 2018;159:61-83.

41. Luu BL, Inglis JT, Hurn TP, Van der Loos HF, Croft EA, Blouin JS. Human standing is modified by an unconscious integration of congruent sensory and motor signals. *The Journal of physiology*. 2012;590(22):5783-5794.
42. Luu BMSFoMU. Perception, perfusion and posture. In: Fitzpatrick RNRA, ed2010.
43. Tisserand R, Rasman BG, Omerovic N, Peters RM, Forbes PA, Blouin J-S. Unperceived motor actions of the balance system interfere with the causal attribution of self-motion. *PNAS Nexus*. 2022;1(4).
44. Gonshor A, Jones GM. Extreme vestibulo-ocular adaptation induced by prolonged optical reversal of vision. *The Journal of physiology*. 1976;256(2):381-414.
45. Jones GM. Plasticity in the adult vestibulo-ocular reflex arc. *Philos Trans R Soc Lond B Biol Sci*. 1977;278(961):319-334.
46. Forbes PA, Luu BL, Van der Loos HF, Croft EA, Inglis JT, Blouin JS. Transformation of Vestibular Signals for the Control of Standing in Humans. *The Journal of neuroscience : the official journal of the Society for Neuroscience*. 2016;36(45):11510-11520.
47. Assaiante C, Amblard B. An ontogenetic model for the sensorimotor organization of balance control in humans. *Human movement science*. 1995;14(1):13-43.
48. Dorfman LJ, Bosley TM. Age-related changes in peripheral and central nerve conduction in man. *Neurology*. 1979;29(1):38-44.
49. Eyre JA, Miller S, Ramesh V. Constancy of central conduction delays during development in man: investigation of motor and somatosensory pathways. *The Journal of physiology*. 1991;434:441-452.
50. Lin SI, Woollacott MH. Postural muscle responses following changing balance threats in young, stable older, and unstable older adults. *J Mot Behav*. 2002;34(1):37-44.
51. Woollacott MH, Shumway-Cook A, Nashner LM. Aging and posture control: changes in sensory organization and muscular coordination. *Int J Aging Hum Dev*. 1986;23(2):97-114.
52. Gurfinkel VS, Levik YS, Popov KE, Smetanin BN, Shlikov VY. Body Scheme in the Control of Postural Activity. In: Gurfinkel VS, Ioffe ME, Massion J, Roll JP, eds. *Stance and Motion: Facts and Concepts*. Boston, MA: Springer US; 1988:185-193.
53. Massion J. Postural control system. *Current opinion in neurobiology*. 1994;4(6):877-887.
54. Berniker M, Franklin DW, Flanagan JR, Wolpert DM, Kording K. Motor learning of novel dynamics is not represented in a single global coordinate system: evaluation of mixed coordinate representations and local learning. *Journal of neurophysiology*. 2014;111(6):1165-1182.
55. Berniker M, Kording K. Estimating the sources of motor errors for adaptation and generalization. *Nature neuroscience*. 2008;11(12):1454-1461.
56. Goodworth AD, Peterka RJ. Influence of stance width on frontal plane postural dynamics and coordination in human balance control. *Journal of neurophysiology*. 2010;104(2):1103-1118.
57. Goodworth AD, Peterka RJ. Sensorimotor integration for multisegmental frontal plane balance control in humans. *Journal of neurophysiology*. 2012;107(1):12-28.
58. Loram ID, Maganaris CN, Lakie M. Human postural sway results from frequent, ballistic bias impulses by soleus and gastrocnemius. *The Journal of physiology*. 2005;564(Pt 1):295-311.
59. Torres-Oviedo G, Ting LH. Subject-specific muscle synergies in human balance control are consistent across different biomechanical contexts. *Journal of neurophysiology*. 2010;103(6):3084-3098.
60. Kiemel T, Zhang Y, Jeka JJ. Identification of neural feedback for upright stance in humans: stabilization rather than sway minimization. *The Journal of neuroscience : the official journal of the Society for Neuroscience*. 2011;31(42):15144-15153.
61. Milton J, Cabrera JL, Ohira T, et al. The time-delayed inverted pendulum: implications for human balance control. *Chaos*. 2009;19(2):026110.

62. Peterka RJ. Sensorimotor integration in human postural control. *Journal of neurophysiology*. 2002;88(3):1097-1118.
63. Fitzpatrick R, Burke D, Gandevia SC. Loop gain of reflexes controlling human standing measured with the use of postural and vestibular disturbances. *Journal of neurophysiology*. 1996;76(6):3994-4008.
64. Fitzpatrick R, Rogers DK, McCloskey DI. Stable human standing with lower-limb muscle afferents providing the only sensory input. *The Journal of physiology*. 1994;480 (Pt 2):395-403.
65. Fukuoka Y, Nagata T, Ishida A, Minamitani H. Characteristics of somatosensory feedback in postural control during standing. *IEEE transactions on neural systems and rehabilitation engineering : a publication of the IEEE Engineering in Medicine and Biology Society*. 2001;9(2):145-153.
66. Loram ID, Gawthrop PJ, Lakie M. The frequency of human, manual adjustments in balancing an inverted pendulum is constrained by intrinsic physiological factors. *The Journal of physiology*. 2006;577(Pt 1):417-432.
67. Loram ID, Kelly SM, Lakie M. Human balancing of an inverted pendulum: is sway size controlled by ankle impedance? *The Journal of physiology*. 2001;532(Pt 3):879-891.
68. Luu BL, Huryn TP, Van der Loos HF, Croft EA, Blouin JS. Validation of a robotic balance system for investigations in the control of human standing balance. *IEEE transactions on neural systems and rehabilitation engineering : a publication of the IEEE Engineering in Medicine and Biology Society*. 2011;19(4):382-390.
69. van der Kooij H, Campbell AD, Carpenter MG. Sampling duration effects on centre of pressure descriptive measures. *Gait & posture*. 2011;34(1):19-24.
70. Day BL, Severac Cauquil A, Bartolomei L, Pastor MA, Lyon IN. Human body-segment tilts induced by galvanic stimulation: a vestibularly driven balance protection mechanism. *The Journal of physiology*. 1997;500 (Pt 3):661-672.
71. Horak FB, Diener HC, Nashner LM. Influence of central set on human postural responses. *Journal of neurophysiology*. 1989;62(4):841-853.
72. Horak FB, Nashner LM. Central programming of postural movements: adaptation to altered support-surface configurations. *Journal of neurophysiology*. 1986;55(6):1369-1381.
73. Hsu WL, Scholz JP, Schoner G, Jeka JJ, Kiemel T. Control and estimation of posture during quiet stance depends on multijoint coordination. *Journal of neurophysiology*. 2007;97(4):3024-3035.
74. Kuo AD, Zajac FE. Human standing posture: multi-joint movement strategies based on biomechanical constraints. *Progress in brain research*. 1993;97:349-358.

REVIEWERS' COMMENTS:

Reviewer #1 (Remarks to the Author):

As the authors have responded correctly to my comments, I recommend acceptance of the paper.

Reviewer #2 (Remarks to the Author):

In this revision, the authors proposed a new analysis of the learning effect they investigate, thus providing some evidence - albeit faint and indirect - of an after-effect in the first experiment. Together with new simulations showing how an optimal controller would display a qualitatively similar signature in the measure used to assess learning, the revised manuscript now better supports the claim that low-level motor learning is actually taking place when participants adapt to delays when standing. Along with a notable effort to make their manuscript clearer and a more thorough discussion, I find this new version much more compelling.

While I cannot judge if all the concerns raised by the first reviewer were properly addressed, I find that my previous comments are now resolved and would therefore recommend proceeding with publication.